# TOONCOMPOSER: STREAMLINING CARTOON PRODUCTION WITH GENERATIVE POST-KEYFRAMING

**Lingen Li**[1,3*]     **Guangzhi Wang**[2†]     **Zhaoyang Zhang**[2†]     **Yaowei Li**[4]     **Xiaoyu Li**[2]
**Qi Dou**[3]     **Jinwei Gu**[3]     **Tianfan Xue**[1,5‡]     **Ying Shan**[2]

[1]CUHK MMLab   [2]ARC Lab, Tencent PCG   [3]CUHK   [4]PKU   [5]CPII under InnoHK

lgli@link.cuhk.edu.hk

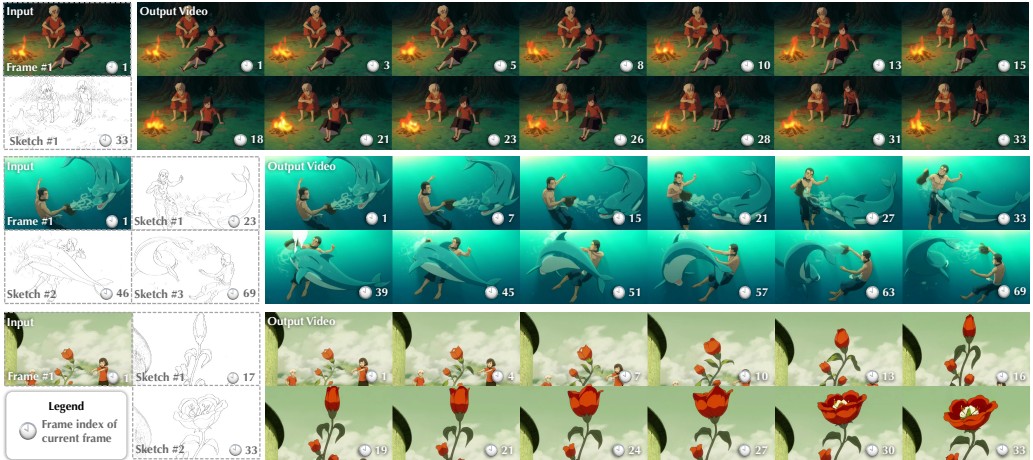

Figure 1: Video samples generated by ToonComposer using sparse keyframe sketches, featuring scenes from cartoon movies (used with permission). ToonComposer supports precise keyframe control and flexible inference with varying numbers of input keyframe sketches and output video lengths (33 frames for the first and third samples, and 69 frames for the second sample). Frames are evenly sampled for illustration. Each input and output frame is annotated with its corresponding temporal index in the bottom right corner. These movies were excluded from the training data.

## ABSTRACT

Traditional cartoon and anime production involves keyframing, inbetweening, and colorization stages, which require intensive manual effort. Despite recent advances in AI, existing methods often handle these stages separately, leading to error accumulation and artifacts. For instance, inbetweening approaches struggle with large motions, while colorization methods require dense per-frame sketches. To address this, we introduce ToonComposer, a generative model that unifies inbetweening and colorization into a single post-keyframing stage. ToonComposer employs a sparse sketch injection mechanism to provide precise control using keyframe sketches. Additionally, we propose a novel cartoon adaptation method with the spatial low-rank adapter to effectively tailor a modern video foundation model to the cartoon domain while keeping its temporal prior intact. Requiring as few as a single sketch and a colored reference frame, ToonComposer excels with sparse inputs, while also supporting multiple sketches at any temporal location for more precise motion control. This dual capability reduces manual workload and improves flexibility, empowering artists in real-world scenarios. To evaluate our model, we further created PKBench, a benchmark featuring human-drawn sketches that simulate real-world use cases. Our evaluation demonstrates that ToonComposer outperforms existing methods in visual quality, motion consistency, and production efficiency, offering a superior and more flexible solution for AI-assisted cartoon production.

---

*Work done during the internship at ARC Lab. Project page: https://lg-li.github.io/project/tooncomposer.
†Project lead.     ‡Corresponding authors.

# 1 INTRODUCTION

Cartoons and anime are celebrated for their vibrant aesthetics and intricate narratives, standing as a cornerstone of global entertainment. Traditional cartoon production involves keyframing, inbetweening, and colorization stages, each of which requires artists to craft numerous frames to ensure fluid motion and stylistic consistency (Tang et al., 2025). While the keyframing stage is a creative process that embodies human artistry, the subsequent inbetweening and colorization stages are highly labor-intensive and time-consuming. Specifically, the inbetweening and colorization stages, which require less creative input, demand the production of hundreds of drawings for mere seconds of animation, resulting in significant time and resource costs.

Recent advances in generative models have facilitated the inbetweening and colorization stages, such as ToonCrafter (Xing et al., 2024a; Jiang et al., 2024), AniDoc (Meng et al., 2024), or LVCD (Huang et al., 2024b). However, these methods face critical limitations: (1) inbetweening approaches (Xing et al., 2024a; Jiang et al., 2024) struggle to interpolate large motions from sparse sketch inputs, often requiring multiple keyframes for a smooth motion; (2) colorization methods (Meng et al., 2024; Huang et al., 2024b) demand detailed per-frame sketches, imposing significant artist workload. (3) Additionally, their sequential processing leads to error accumulation, where inaccuracies in interpolated sketches affect the colorization stage, resulting in artifacts and reduced quality (Tang et al., 2025). These shortcomings highlight a significant gap in achieving a streamlined and efficient production pipeline that produces high-quality results.

In fact, the inbetweening and colorization stages are highly interdependent. Both of them require searching for correspondences among keyframe sketches or color reference frames. Therefore, we introduce the **post-keyframing** stage, a novel paradigm that follows the keyframe creation stage and merges inbetweening and colorization into a single automated process. This unification enables the model to jointly utilize the information in keyframe sketches and color reference frames in a single stage, avoiding the risk of cross-stage error accumulation. As illustrated in Figure 2, the post-keyframing stage requires only a few keyframe sketches and a colored reference frame to generate a complete high-quality cartoon video. This approach significantly reduces manual effort, allowing artists to focus on creative keyframe design, while AI manages repetitive tasks.

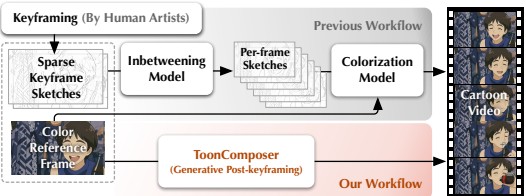

Figure 2: Comparison between previous cartoon production workflow and ours. ToonComposer enables the *post-keyframing* stage, seamlessly integrating inbetweening and colorization into a single automated process, streamlining cartoon production compared to previous traditional and existing AI-assisted workflows.

To realize the full potential of post-keyframing, we leverage the rich generative priors of modern video foundation models (Wang et al., 2025a). However, adapting these models for the post-keyframing stage presents unique challenges: 1) *Precise sketch controllability*: Foundation models are typically designed for weak conditioning via text prompts or initial frames. They lack the inherent mechanism to incorporate *sparse* keyframe sketches that require precise alignment at specific temporal positions, creating a gap between the model's native capabilities and the granular control required by artists; 2) *Cartoon adaptation*: While domain adaptation is essential for cartoon generation, the architecture differences between video foundation models pose a significant hurdle. Previous successes in cartoon adaptation (Xing et al., 2024a) relied on the explicit structural decoupling of spatial and temporal layers in UNet-based models, allowing for spatial-only tuning by simply freezing temporal layers. In contrast, modern Diffusion Transformers (DiT) employ *fully coupled* spatio-temporal attention structures. This coupling renders previous layer-wise freezing strategies inapplicable, as standard adaptation methods may inevitably disrupt the pretrained temporal priors.

To address these challenges, we propose **ToonComposer**, a specialized generative model for post-keyframing. First, to bridge the controllability gap, we devise a *sparse sketch injection* mechanism, which seamlessly integrates sparse keyframe sketches into the token sequence and enables precise temporal keyframe control. Second, to resolve the spatial adaptation in spatio-temporal-coupled structures, we introduce the Spatial Low-Rank Adapter (SLRA). Unlike standard adapters, SLRA is designed for spatial-temporal-coupled modules in models like DiT, which constrains adaptation to the spatial scope. This effectively tailors the appearance of the foundation model to the cartoon

Figure 3: Model design of ToonComposer. A sparse sketch injection mechanism enables precise control using keyframe sketches, and a cartoon adaptation method incorporating a spatial low-rank adapter tailors the DiT-based video model to the cartoon domain, preserving its temporal priors.

domain while keeping the temporal motion prior intact. In addition, we enhance the model flexibility with *region-wise control*, empowering artists to guide generation in specific areas while leaving others for the model to generate based on context. These techniques ensure that ToonComposer generates high-quality cartoon videos with minimal input, as shown in Figure 1, effectively establishing a streamlined post-keyframing paradigm for cartoon production.

To support the training of the proposed model, we curated a dataset PKData, which contains high-quality anime and cartoon video clips. Each clip is accompanied by keyframe sketches in multiple versions, enhancing the robustness of the model to different sketch styles. In addition to evaluating our model on benchmark with synthetic sketches, we curated PKBench, a new benchmark that contains 30 original cartoon scenes with human-drawn keyframe sketches and reference color frames. Extensive experiments on both benchmarks demonstrate that ToonComposer outperforms existing methods in visual quality, motion coherence, and production efficiency. Our contributions are summarized as follows:

- We propose ToonComposer, a new generative model that establishes the post-keyframing paradigm. It features a specialized sparse sketch injection mechanism to enable precise control with minimal input.

- We design a cartoon adaptation mechanism using SLRA, which effectively tailors the spatial behavior of a spatio-temporal-coupled video generation model to the cartoon domain while preserving its temporal prior.

- We curate a cartoon post-keyframing dataset with diverse sketches for training and develop PKBench featuring human-drawn sketches, facilitating the evaluation in real cartoon production scenarios.

- Extensive experiments on both synthetic and human-drawn benchmarks demonstrated the superiority of our method, with additional ablation studies validated the effectiveness of each proposed component in our framework.

## 2 RELATED WORK

**AI-assisted Cartoon Production**   AI has increasingly been applied to automate labor-intensive tasks in cartoon and anime production (Tang et al., 2025; Zhang et al., 2025a; Yang et al., 2025b), such as inbetweening and colorization. For inbetweening, early methods like AnimeInterp (Li et al., 2021) and AutoFI (Shen et al., 2022) focus on linear and simple motion interpolation. More recently, diffusion-based methods (Xing et al., 2024a; Jiang et al., 2024) become capable of handling cases with more complex motion by harnessing the generative priors of a pretrained model. For colorization, early GAN-based (Isola et al., 2017) and recent diffusion-based methods (Zhuang et al., 2024; Meng et al., 2024; Huang et al., 2024b; Zhuang et al., 2025; Chen et al., 2025; Zhang et al., 2025c; Sadihin et al., 2025) have automated the colorization of line art based on one or a series of reference frames. Recent work also adopts layer-wise control for cartoon generation (Yang et al., 2025b). However, while existing AI-assisted methods have advanced cartoon production by automating inbetweening and colorization, they require dense frame inputs or operate as separate, isolated stages, facing challenges with complex motions and stylistic consistency (Tang et al., 2025). ToonComposer overcomes these hurdles by offering a unified, sparse-input solution for post-keyframing that simplifies the production workflow.

**Video Diffusion Model**   Diffusion models have emerged as the cornerstone for generative tasks (Ho et al., 2020), particularly in image and video synthesis, by iteratively denoising samples from a noise distribution to produce high-quality outputs (Blattmann et al., 2023b). For video

generation, these models must effectively capture both spatial details and temporal dynamics, a challenge that has led to distinct architectural approaches. Traditional UNet-based diffusion models (Ho et al., 2022; Blattmann et al., 2023b;a; Xing et al., 2024c) extend 2D U-Nets to handle videos by incorporating 3D convolutions and separated spatial and temporal attention layers. In these models, spatial attention layers process intra-frame features, often across channels or spatial positions, while separate temporal attention layers model inter-frame dependencies. In contrast, Diffusion Transformers (DiTs) (Peebles & Xie, 2023) leverage transformer architectures, replacing UNet's convolutional backbone with full attention mechanisms that model long-range dependencies in both spatial and temporal dimensions (Yang et al., 2024; Kong et al., 2024; Wang et al., 2025a). Although showing stronger performance compared to spatial-temporal decoupled design, such full attention mechanism eliminates the availability of spatial adaptation tailored for domains such as cartoon (Xing et al., 2024a). Our work builds upon the DiT-based foundation model to harness the high-quality video prior with a new cartoon adaptation mechanism, which adapts the DiT-based foundation model to the cartoon domain in spatial behavior while keeping its temporal motion prior intact.

**Controllable Generation** Controllable generation aims to steer image and video synthesis (He et al., 2024; Wang et al., 2024b) or editing (Yang et al., 2023; Jiang et al., 2025; Zhu et al., 2025) with explicit conditions such as reference images (Ye et al., 2023; Zhang & Agrawala, 2023; Li et al., 2025a; Zhu et al., 2025; Li et al., 2025b), depth maps (Xing et al., 2024b), human poses (Zhu et al., 2024; Hu, 2024), videos (Jiang et al., 2025; Zhang et al., 2025b; Yang et al., 2025a), etc. Techniques such as IP-Adapter (Ye et al., 2023) and ControlNet (Zhang & Agrawala, 2023) inject these visual cues into diffusion models alongside text prompts, allowing fine-grained manipulation of both content and style. The value of controllability of generation is particularly evident in domain-specific pipelines. For cinematography, camera-aware generators expose handles for 2D scene layout and 3D camera trajectories (He et al., 2024; Wang et al., 2024b; Li et al., 2025c; Wang et al., 2025b), allowing video creators to frame shots and motion with high precision. In cartoon production, sketch-guided models support interpolation, inbetweening, and colorization (Meng et al., 2024; Huang et al., 2024b; Xing et al., 2024a; Zhang et al., 2025c). Our method focuses on the controllable cartoon generation using sparse keyframe sketches with DiT models.

## 3 METHODOLOGY

We introduce ToonComposer, a novel generative post-keyframing model that produces high-quality cartoon videos with sparse control. To achieve this, we propose a curated sparse sketch injection strategy, which effectively enables precise sketch control at arbitrary timestamps (Section 3.2). Furthermore, to fully leverage the temporal prior in video generation models, we design a novel low-rank adaptation strategy that efficiently adapts the spatial prior to the cartoon domain while leaving the temporal prior intact (Section 3.3). To further alleviate artist workload and improve efficiency, our method also enables region-wise control, empowering artists to draw only part of sketches while leaving the model to reason how the motion should be generated in blank areas (Section 3.4).

### 3.1 POST-KEYFRAMING STAGE

The cartoon industry has benefited significantly from the development of generative AI, facilitating the stage of inbetweening (Xing et al., 2024a) and colorization (Huang et al., 2024b; Meng et al., 2024). The two stages are highly interdependent: both require searching and interpolating along the correspondence between elements in the keyframes/sketches, indicating that their internal mechanisms are similar. Merging the two processes significantly alleviates the requirement of dense per-frame sketches and avoids the risk of cross-stage error accumulation. Motivated by this, we propose the *post-keyframing* stage, a new stage that automates cartoon production and consolidates the inbetweening and colorization into a unified generative process. Given **one** colored reference frame and **one** sketch frame, the *post-keyframing* stage aims to directly produce a high-quality cartoon video that adheres to the guidance provided by these inputs.

Formally, given a colored reference frame $f_1$ and a sketch frame $s_j$, we aim to obtain a model $\mathbf{G}_\theta$ that directly generates a high-quality cartoon video with $K$ frames:

$$\{\hat{f}_k\}_{k=1}^K = \mathbf{G}_\theta(f_1, s_j, e_{\text{text}}), \tag{1}$$

where $j$ represents the temporal location of $s_j$, and $e_{\text{text}}$ represents the prompt describing the scene.

## 3.2 SPARSE SKETCH INJECTION

Advanced video generation models, such as Wan (Wang et al., 2025a), demonstrate exceptional performance in producing high-quality videos. While image-to-video (I2V) video generation models variant supports video generation guided by an initial frame, precise control using sparse sketches at arbitrary temporal positions remains unexplored. To this end, we introduce a novel *sparse sketch injection* mechanism that integrates sketches into the latent token sequence of an I2V DiT model for precise temporal control.

In the standard I2V DiT model $\epsilon_\theta$, the input image is concatenated with the noisy latent $z$ along the channel dimension. To inject the sketch frame $s_j$ into the latent representation of the DiT model $\epsilon_\theta$ for precise control over the temporal location $j$ in the generated cartoon, we first introduce an additional projection head that embeds the conditional sketch latents into sketch tokens $s_j'$ that are compatible with the latent dimension of the model. Then, we apply the *position embedding mapping* process that borrows the RoPE (Su et al., 2024) encodings from the corresponding video tokens at the temporal index $j$ before each attention operator. These sparse sketch tokens are concatenated with the video tokens along the sequence dimension to facilitate the attention computation process.

This mechanism enables efficient integration of sketch conditions into the latent space with temporal awareness during the generation process. In addition, it facilitates the simultaneous use of multiple keyframes and sketches as control input. Given the complexity of motion in some cartoon scenes, precise control often necessitates multiple keyframes and sketches. Therefore, we extend the formulation to support both multiple colored reference frames and multiple sketch inputs. Consequently, the forward step of the I2V DiT model with sparse sketch injection is expressed as:

$$\hat{\epsilon} = \epsilon_\theta \left( \left[ [\{z_k^{(t)}\}_{k=1}^K, \mathrm{pad}(\{f_{i_c}'\}_{c=1}^C)]_c, \{s_{i_n}'\}_{n=1}^N \right]_s, e_{\text{text}}, t \right), \qquad (2)$$

where $\{s_{i_n}'\}_{n=1}^N$ represents $N$ sketch frames and $\{f_{i_c}'\}_{c=1}^C$ denotes $C$ colored reference frames. $[\cdot, \cdot]_s$ means concatenation along the token sequence dimension. This formulation enables precise control over multiple inputs, while also supporting the minimal input requirement of the *post-keyframing* stage (one colored and one sketch frame, described in Section 3.1).

In addition, an extra position-aware residual is introduced in our model to enhance the flexibility of sketch control. Please refer to the Appendix Section A.1 for more details.

## 3.3 CARTOON ADAPTATION

Previous work (Xing et al., 2024a) has demonstrated the success of adapting video generation models to the cartoon domain by tuning only the spatial layers of a spatial-temporal U-Net, preserving the temporal motion prior while adapting appearance. However, modern video models (Yang et al., 2024; Wang et al., 2025a; Kong et al., 2024) employ 3D-full attention, intertwining spatial and temporal representations, making direct spatial adaptation infeasible.

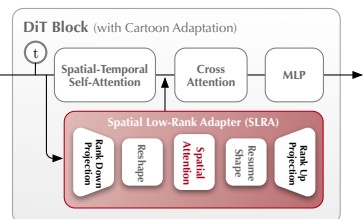

Figure 4: SLRA used for cartoon adaptation. It takes the hidden states before the spatial-temporal self-attention module as input and outputs a residual that is added after the self-attention operation.

To address this, we introduce the Spatial Low-Rank Adapter (SLRA), a novel low-rank adaptation mechanism that modifies the attention module's spatial behavior while preserving the temporal prior. As shown Figure 4, given a token sequence $h \in \mathbb{R}^{L \times D}$ in each self-attention module of $\epsilon_\theta$, SLRA produces residual $h_{\text{res}}$ to adapt the original self-attention modules in the DiT model for the cartoon domain:

$$h_{\text{res}} = [\underset{\tilde{\mathbf{W}}}{attn}([h\mathbf{W}^{\text{down}}]_{\text{reshape}})\mathbf{W}^{\text{up}}]_{\text{resume}}, \qquad (3)$$

where $attn(\cdot)$ performs self-attention independently on the spatial dimension of each frame, with the same positional embeddings as the main model applied to the video and sketch tokens. $\mathbf{W}^{\text{down}} \in \mathbb{R}^{D \times D'}$ and $\mathbf{W}^{\text{up}} \in \mathbb{R}^{D' \times D}$ are trainable downsampling and upsampling matrices operating on the feature dimension. $\tilde{\mathbf{W}} = \{\mathbf{W}_Q, \mathbf{W}_K, \mathbf{W}_V, \mathbf{W}_O\}$, where $\mathbf{W}_Q, \mathbf{W}_K, \mathbf{W}_V, \mathbf{W}_O \in \mathbb{R}^{D' \times D'}$ are the trainable matrices in the SLRA's attention. Assume $H$ and $W$ are the spatial sizes of DiT's latent tokens, and $K$ and $N$ are the temporal length of the video tokens and sketch tokens. The reshape operation $[\cdot]_{\text{reshape}}$ reorganizes the spatial-temporal token arrangement into $\mathbb{R}^{(K+N) \times (H \times W) \times D'}$,

where $D' \ll D$, constraining attention to the spatial dimension ($H \times W$) only. The resume-shape operation $[\cdot]_{\text{resume}}$ restores the sequence form to $\mathbb{R}^{L \times D'}$. Distinct from vanilla LoRA, SLRA explicitly restricts information propagation to the spatial dimension, leaving the temporal dimension intact. This effectively adapts the appearance of the base video generation model to the cartoon domain, while maintaining its strong temporal prior. As demonstrated in Table 3, SLRA outperforms LoRA and its variants. Please refer to Appendix Section A.2 for more details.

### 3.4 REGION-WISE CONTROL

Sometimes cartoon creators may only want to draw the foreground sketch and let the generator create the background for them. If they simply leave the background blank, this may result in undesirable artifacts, as shown in the second row of Figure 7.

To this end, we propose a novel region-wise control mechanism that allows artists to specify blank regions in sketches for the model to generate plausible content based on context or text prompts. During training, random masks $m_{i_n} \in \{0,1\}^{H \times W}$ are applied to the sketch frames $s_{i_n}$, where $m_{i_n}(i, j) = 0$ indicates a region where the sketch is not provided. An additional channel is concatenated to $s_{i_n}$, which is encoded as:

$$\tilde{s}'_{i_n} = [\mathcal{E}(s_{i_n}), m_{i_n}]_{\text{c}}, \tag{4}$$

where $\tilde{s}'_{i_n}$ is used to replace the $s'_{i_n}$ described in Equation (2) during training. The model learns to reconstruct full frames in masked regions, enabling flexible inference where artists can assign the value of $m_{i_n}$ and leave masked areas blank for context-driven generation.

Complementary to the support of temporally sparse keyframes and sketches, our region-wise control allows the input sketch to be spatially sparse, further alleviating the requirements and labors for cartoon creators.

### 3.5 TRAINING OBJECTIVE

ToonComposer is trained as a conditional diffusion model following Rectified Flow (Esser et al., 2024), which predicts the velocity $v_t$ at a timestep $t$ sampled from logit-normal distribution. For simplicity, we write the input part in Equation (2) as $x_{\text{in}}$:

$$x_{\text{in}} = \left[ [\{z_k^{(t)}\}_{k=1}^K, \text{pad}(\{f'_{i_c}\}_{c=1}^C)]_{\text{c}}, \{\tilde{s}'_{i_n}\}_{n=1}^N \right]_{\text{s}}, \tag{5}$$

and let $\mathbf{z}_0 = \{z_k^{(0)}\}_{k=1}^K$ be a clean cartoon video latent, the training objective minimizes the expected velocity prediction error:

$$\mathcal{L} = \mathbb{E}_{\mathbf{z}_0, \eta, t} \left[ \|v_t - \epsilon_\theta(x_{\text{in}}, e_{\text{text}}, t)\|_2^2 \right], \tag{6}$$

where $\eta$ is the random Gaussian noise, $v_t$ is the velocity derived from $\{z_k^{(t)}\}_{k=1}^K - \eta$, and $\epsilon_\theta$ is the ToonComposer model to be trained.

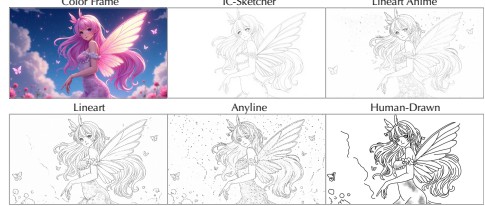

## 4 EXPERIMENTS

### 4.1 EXPERIMENTAL SETTINGS

**Dataset** Based on web video sources, we filtered and constructed the PKData, a high-quality cartoon dataset containing 37K diverse cartoon video clips. Each clip was accompanied by a descriptive caption generated by CogVLM (Wang et al., 2024a) and a set of sketch frames. Recognizing the stylistic diversity in sketches due to different artist preferences or creation tools, we augmented our dataset with diverse types of sketches. Specifically, we synthesize four versions of sketches per frame using four open-source CNN-based sketch models, including two basic lineart models used in ControlNet (Zhang & Agrawala, 2023), Anime2Sketch (Xiang et al.,

Figure 5: Examples of different sketch types used during training and evaluation. All variants except human-drawn sketches are included in the training set. The diversity of training sketches improves ToonComposer's robustness to varying sketch styles in real use cases. Human-drawn sketches are used for evaluation, as discussed in Section 4.3.

Table 1: Quantitative evaluation results on the synthetic benchmark, comparing ToonComposer with previous AI-assisted cartoon generation methods: AniDoc (Meng et al., 2024), LVCD (Huang et al., 2024b), and ToonCrafter (Xing et al., 2024a).

| Method | LPIPS↓ | DISTS↓ | CLIP↑ | Subject Con.↑ | Motion Smo.↑ | Background Con.↑ | Aesthetic Qua.↑ |
|---|---|---|---|---|---|---|---|
| AniDoc | 0.3734 | 0.5461 | 0.8665 | 0.9067 | 0.9798 | 0.9408 | 0.4962 |
| LVCD | 0.3910 | 0.5505 | 0.8428 | 0.8316 | 0.9810 | 0.9183 | 0.4984 |
| ToonCrafter | 0.3830 | 0.5571 | 0.8463 | 0.8075 | 0.9550 | 0.8920 | 0.5035 |
| ToonComposer (1.3B) | **0.1698** | 0.1097 | 0.9292 | 0.9243 | **0.9889** | 0.9505 | 0.5576 |
| ToonComposer (14B) | 0.1785 | **0.0926** | **0.9449** | **0.9451** | 0.9886 | **0.9547** | **0.5999** |

2022), and Anyline (Soria et al., 2023). Furthermore, we tune an image-to-image generative model from FLUX.1-dev with in-context LoRA (Huang et al., 2024a) on a small real-sketch dataset from multiple artists. This model, named IC-Sketcher, is then used to produce another version of sketches. Figure 5 illustrates one example frame with diverse sketches.

**Benchmark** We first evaluate our methods on a synthetic benchmark obtained from cartoon movies (use with permission, for evaluation only), where sketches for each video frame are produced using sketch models. We adopt reference-based metrics on this benchmark since the ground truth is available. Furthermore, we developed *PKBench*, a novel benchmark featuring human-drawn sketches to enable a more comprehensive evaluation of cartoon post-keyframing in real-world scenarios. PKBench contains 30 samples, each including 1) a colored reference frame, 2) a textual prompt that describes the scene, and 3) two real sketches that depict the start and end frames of a scene, drawn by professional artists.

**Metrics** For evaluation metrics, we adopt 1) reference-based perceptual metrics for synthetic benchmark, including LPIPS (Zhang et al., 2018), DISTS (Ding et al., 2020), and CLIP (Radford et al., 2021) image similarity, 2) reference-free video quality metrics from VBench (Huang et al., 2024c) for both synthetic and real benchmarks, including subject consistency (S. C.), motion consistency (M. C.), background consistency (B. C.) and aesthetic quality (A. Q.). 3) A user study on human perceptual quality for the real benchmark.

**Training Details** We adopt Wan 2.1 (Wang et al., 2025a) (both 14B and 1.3B) as our base model and apply the injection and adaptation techniques outlined in Section 3. The model is then trained on our dataset for 10 epochs with an effective batch size of 16, using the AdamW optimizer (Loshchilov & Hutter, 2017) and a learning rate of $10^{-5}$. The SLRA internal dimension $D'$ is set to 144 by default. We use the zero redundancy optimizer (Rajbhandari et al., 2020) stage 2 to reduce memory cost during training.

## 4.2 EVALUATION ON SYNTHETIC BENCHMARK

We first evaluate our ToonComposer on a synthetic cartoon benchmark and compare it with previous methods, including AniDoc (Meng et al., 2024), LVCD (Huang et al., 2024b), and Toon-Crafter (Xing et al., 2024a). In this synthetic evaluation, sketches are obtained from cartoon video frames using the same sketch model (Xiang et al., 2022). To ensure evaluation fairness, we align the ground truths in both spatial and temporal dimension to fit the pre-defined settings of each model for metrics calculation.

**Baseline Methods** Although our model requires only one inference to get the final cartoon video, previous methods demand a two-stage process, as shown in Figure 2. For ToonCrafter (Xing et al., 2024a), we first generate the dense sketch sequence by interpolating the first and last sketch frames, then we use its sketch guidance mode (which requires the first and last color frames as input) to generate the final cartoon video. For LVCD (Huang et al., 2024b) and AniDoc (Meng et al., 2024), we first generate the dense sketch sequence interpolated by ToonCrafter, then colorize the sketches into a final cartoon video using the two models, respectively.

**Results** Table 1 shows the numeric results of the synthetic evaluation. Our method (both the 14B and 1.3B variants) outperforms previous methods in both reference-based metrics and reference-free metrics. For example, our model reports a significantly lower DISTS score, indicating that its perceptual quality is much better than that of its counterparts. Figure 6 visualized the qualitative comparison between these methods, with the ground truth video provided as references. In both samples, our method produces smooth and natural cartoon video frames, while other methods fail to handle such challenging cases with sparse sketches.

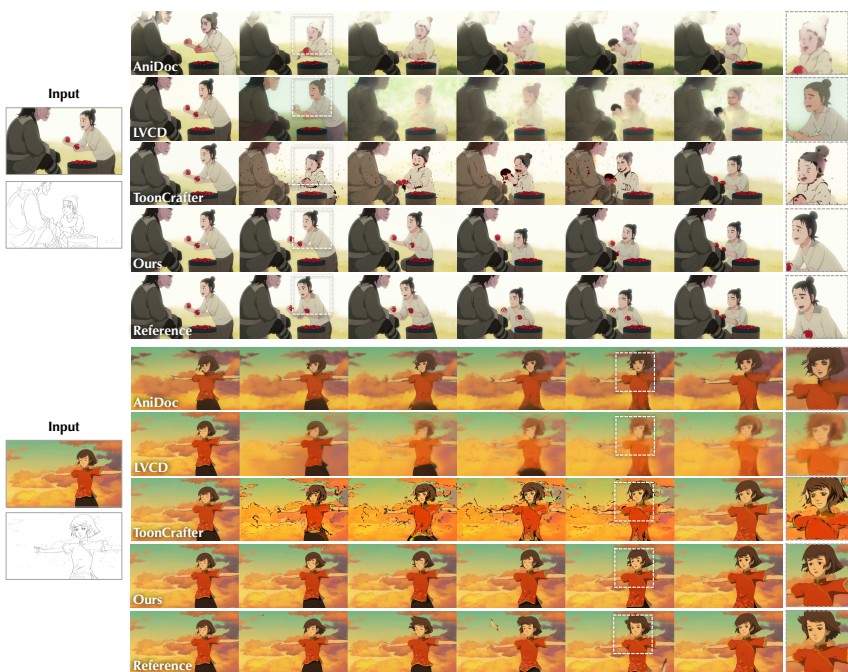

Figure 6: Comparison on the synthetic benchmark among AniDoc, LVCD, ToonCrafter, and our ToonComposer (two variants). Zoom-in patches of a randomly selected region are shown in the rightmost column. Our method demonstrates superior visual quality, smoother motion, and better style consistency with the input image. Evaluation scenes are sourced from movies with permission (*Mr. Miao* and *Big Fish & Begonia*). Please refer to the ***supplementary video*** for more results.

For example, in the zoom-in patches of the first sample, AniDoc and ToonCrafter produce distorted faces. LVCD generates a reasonable face but loses all details in subsequent frames. In contrast, our method produces a clear face which preserves the identity of the first reference frame. These observations align with our method's numeric performance advantages in Table 1. More results are provided in the ***supplementary video***.

Table 2: Quantitative evaluation results on the real sketch benchmark PKBench, comparing ToonComposer with other AI-assisted cartoon generation models.

| Method | S. C.↑ | M. S.↑ | B. C.↑ | A. Q.↑ |
|---|---|---|---|---|
| AniDoc | 0.9456 | 0.9842 | 0.9664 | 0.6611 |
| LVCD | 0.8653 | 0.9724 | 0.9394 | 0.6479 |
| ToonCrafter | 0.8567 | 0.9674 | 0.9343 | 0.6822 |
| ToonComposer (1.3B) | 0.9465 | 0.9871 | 0.9653 | 0.7332 |
| ToonComposer (14B) | **0.9509** | **0.9910** | **0.9681** | **0.7345** |

### 4.3 EVALUATION ON REAL BENCHMARK

In addition to the evaluation on the synthetic test set, we further compared all methods on our proposed benchmark PKBench with real human sketches. Since ground truth is not available for each sample, we evaluated the generated videos using reference-free metrics from VBench (Huang et al., 2024c). These sketches are drawn by human artists and are different from algorithmic sketches synthesized by sketches. This is intentionally an out-of-distribution test to evaluate the model performance in real use scenarios. The quantitative comparison is shown in Table 2, where our model outperforms previous methods in all metrics, achieving superior appearance and motion quality. We present more visual comparison results for the real benchmark in Appendix Section B.3 and a human evaluation in Appendix Section B.2.

### 4.4 DISCUSSION AND ANALYSIS

In this section, we discuss key analyses to validate ToonComposer's effectiveness. Due to space limits, more comparisons, ablation studies, and analysis are detailed in the Appendix Section C.

**Use Case of Region-wise Control** We visualize how region-wise control affects the generated video. Without region-wise control, leaving a blank area in the keyframe sketch causes the model to interpret it as a textureless region, resulting in flat areas in the generated frames, as illustrated in the second row of Figure 7. In contrast, with the region-wise control enabled, users can simply draw an area with brush tools to indicate areas that require generating proper motion according to the context. As shown in the last row of Figure 7, our model is able to infer from the input keyframe,

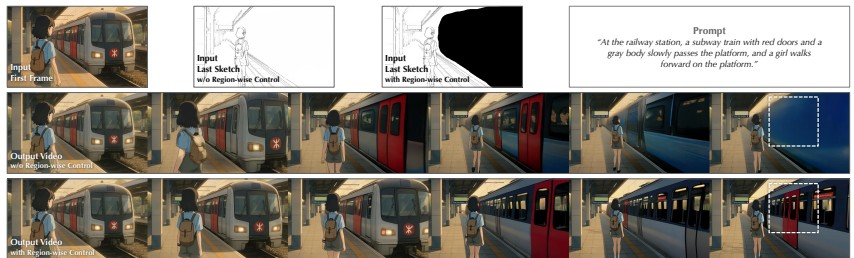

Figure 7: Illustration of region-wise control in ToonComposer. Without region-wise control, blank areas in keyframe sketches are misinterpreted as textureless regions, producing a flat blue train (second row, highlighted with a dashed box). With region-wise control, users can specify areas for context-driven generation without explicit sketches, enabling the model to create plausible and detailed content, such as the dynamic train motion (third row, highlighted with a dashed box).

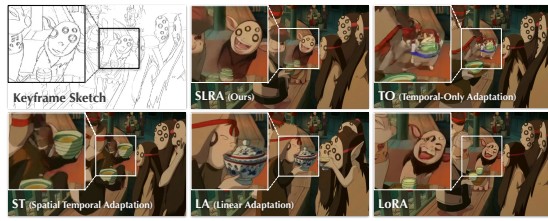

Figure 8: Visualized comparison between the SLRA and alternative adaptation methods. SLRA yields higher visual quality and better coherence with the input keyframe sketches.

Table 3: Ablation study on the SLRA for cartoon adaptation: temporal-only (TO), spatial-temporal (ST), degraded linear adaptation (LA), and the LoRA model. Mixed means both spatial and temporal adaptation are applied.

| Method | Type | Attn. | LPIPS↓ | DISTS↓ | CLIP↑ |
|--------|------|-------|--------|--------|-------|
| SLRA | Spatial | ✔ | **0.1874** | **0.0955** | **0.9634** |
| TO | Temp. | ✔ | 0.1956 | 0.1109 | 0.9581 |
| ST | Mixed | ✔ | 0.1977 | 0.1068 | 0.9587 |
| LA | Mixed | ✗ | 0.2030 | 0.1091 | 0.9589 |
| LoRA | Mixed | ✗ | 0.1922 | 0.1082 | 0.9628 |

the sketch, and the mask given, and automatically generate a plausible movement of the train in the masked area. This mechanism significantly improves flexibility, further alleviating manual workload in real scenarios.

**Ablation on the SLRA** To evaluate the significance of spatial adaptation in ToonComposer, we conducted an ablation study on the SLRA by comparing it against alternative adaptation strategies. These include: **TO** (temporal-only adaptation), which constrains SLRA's internal attention to the temporal dimension, emphasizing temporal dynamics; **ST** (spatial-temporal adaptation), which permits mixed spatial-temporal interactions within the SLRA's attention mechanism; **LA** (linear adaptation), a simplified variant that removes SLRA's attention block entirely, acting as a learnable residual across the whole original attention module; and **LoRA** (Hu et al., 2022), a baseline that applies residuals to all linear layers (query, key, value, and output) in the DiT's attention modules, implicitly affecting both spatial and temporal behaviors. To ensure a fair comparison, we set LoRA's rank to 24 to match SLRA's parameter count, and all models were trained under identical conditions. The results are presented in Table 3 and Figure 8, where SLRA outperforms all variants in both numeric results and visual quality. In Table 3, we denote the adaptation type in *Type* column and whether the adaptation method uses attention operation in *Attn.* column. Specifically, TO (temporal-only adaptation) and ST (spatial-temporal adaptation) yield higher errors due to insufficient or conflicting spatial adjustments, while LA (linear adaptation) lacks the nuanced adaptation required for cartoon aesthetics. Despite the broader scope of LoRA, it underperforms SLRA due to its less targeted adaptation, which disrupts the temporal priors critical for a smooth transition. These findings underscore SLRA's effectiveness in adapting DiT's spatial behavior for cartoon-specific features, while preserving the temporal prior intact.

**Ablation on Sparse Sketch Injection** This mechanism is designed to seamlessly integrate sparse keyframe sketches into the latent token sequence of the DiT-based model while enabling precise temporal keyframe sketch control. To evaluate its effectiveness, we replace it with a conventional channel-wise concatenation approach. As shown in Table 4, this substitution leads to a noticeable degradation in all metrics. This decline underscores the superiority of our sparse sketch injection design. Since we adapt an existing foundation model to the cartoon domain, channel-wise concatenation alters the original structure of noisy latents, increasing the risk of disrupting the pretrained video prior. In contrast, our sparse sketch injection mechanism appends conditional sketches as additional tokens following the original noisy latents and maps corresponding positional embeddings

Table 4: Ablation study on modules: sparse sketch injection (Sparse Sketch Inj.), positional-encoding mapping (Pos. Mapping), and position-aware residual (Position-aware Res.).

| Sparse Sketch Inj. | Pos. Mapping | Position-aware Res. | LPIPS↓ | DISTS↓ | CLIP↑ |
|:---:|:---:|:---:|:---:|:---:|:---:|
| ✔ | ✔ | ✔ | **0.1874** | **0.0955** | **0.9634** |
| ✗ | ✔ | ✔ | 0.2534 | 0.1398 | 0.9493 |
| ✔ | ✗ | ✔ | 0.2893 | 0.1659 | 0.9286 |
| ✔ | ✔ | ✗ | 0.2293 | 0.1097 | 0.9308 |

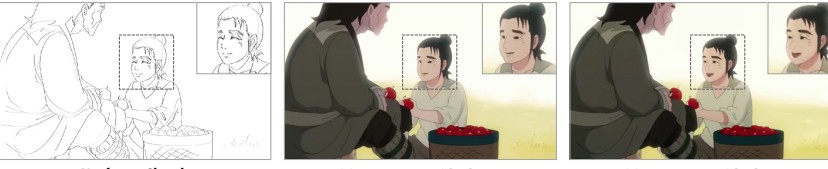

Figure 9: The position-aware residual in ToonComposer includes an adjustable parameter $\alpha$ that users can tune during inference to control the strength of keyframe sketch guidance. Decreasing $\alpha$ in Equation (7) from 1.0 to 0.5 allows for slight deviations from the input sketch (e.g., the boy's mouth shape), while ensuring the generated content remains natural and coherent.

to indicate the specific frame indices the tokens reference, thereby minimizing interference with the original model prior.

**Ablation on Positional-encoding Mapping** The positional-encoding mapping in our sparse sketch injection mechanism leverages RoPE encodings to endow sketch tokens with temporal awareness, ensuring precise alignment of control signals within the video sequence. To validate its effectiveness, we trained a model with the sparse sketch injection mechanism but without the positional-encoding mapping. As shown in Table 4, omitting this module disrupts the model's ability to associate sketches with specific timestamps, leading to significant performance degradation. These results validate the necessity of positional awareness for handling sparse inputs at arbitrary temporal locations, as it prevents misalignment and enhances the model's capacity to generate temporally consistent animations.

**Ablation on Position-aware Residual** This component refines the integration of sketch tokens by adding a scaled residual to the corresponding video tokens, improving conditioning during training, and offering flexibility during inference through an adjustable parameter $\alpha$. Specifically, it adds internal features of the sketch tokens scaled by $\alpha$ through a trainable linear layer to features of the video tokens at keyframe positions. Details of this design are described in Section A.1. According to the results in Table 4, the removal of it causes performance degradation, indicating its role in strengthening the sketch guidance. During inference, the adjustable $\alpha$ allows users to fine-tune the control strength of keyframe sketches. For instance, as illustrated in Figure 9, reducing $\alpha$ from the default value of 1.0 to 0.5 relaxes adherence to the sketch, allowing subtle deviations (e.g., the boy's mouth shape) while preserving overall plausibility and coherence. This adaptability makes ToonComposer more versatile for creative workflows, where varying degrees of control may be desired.

Collectively, these ablation studies above affirm that the proposed modules are integral to ToonComposer's effectiveness, enabling robust handling of sparse inputs and high-quality cartoon generation.

## 5 CONCLUSION

In this paper, we present ToonComposer, a novel model that streamlines cartoon production by automating tedious tasks of inbetweening and colorization through a unified generative process named *post-keyframing*. Built on the DiT architecture, ToonComposer leverages sparse keyframe sketches and a single colored reference frame to produce high-quality, stylistically consistent cartoon video sequences. Experiments show that ToonComposer surpasses existing methods in visual fidelity, motion coherence, and production efficiency. Features such as sparse sketch injection and region-wise control empower artists with precision and flexibility, making ToonComposer a versatile system for cartoon creation. Despite limitations such as computational costs, ToonComposer offers a promising solution to streamline the cartoon production pipeline through generative models.

## ACKNOWLEDGMENTS

This study was supported by CUHK-CUHK(SZ)-GDSTC Joint Collaboration Fund No. 2025A0505000053. The authors thank the reviewers for their insightful comments.

## ETHICS STATEMENT

The authors have read and adhere to the ICLR Code of Ethics. This work focuses on advancing generative models for cartoon production, which involves no real human subjects, sensitive data, or potentially harmful applications. There are no concerns regarding discrimination, bias, fairness, privacy, security, legal compliance, or research integrity. The authors confirm no conflicts of interest or sponsorship issues that could compromise the work. Datasets are used with permission.

## REPRODUCIBILITY STATEMENT

To facilitate reproducibility, the authors provide detailed descriptions of our model architecture, training procedures, and evaluation metrics in the main paper and appendix. Codes and weights are available at `https://github.com/TencentARC/ToonComposer` for reproducibility and community use.

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

CONTENTS

## A MODEL DESIGN DETAILS

Due to the space limit in the main paper, this section provides a detailed explanation for the model designs of the position-aware residual and the SLRA in ToonComposer.

### A.1 POSITION-AWARE RESIDUAL

To enhance the flexibility of sketch control, our ToonComposer also allows users to dynamically adjust the control strength of input sketches, through an adjustable weight during inference. This is done through a *positional-aware residual* module in the sparse sketch injection process. For sparse sketch tokens at controlled keyframe indices $\{i_n\}_{n=1}^N$, we transform the features of sketch tokens inside a DiT block through a linear layer $\mathbf{W}_{\text{res}}$ and combine them with the corresponding features of video tokens of current DiT block at matching indices, scaled by a weight $\alpha \in [0, 1]$ :

$$\{\tilde{z}_k^{(t)}\}_{k \in \{i_n\}_{n=1}^N} := \{\tilde{z}_k^{(t)}\}_{k \in \{i_n\}_{n=1}^N} + \alpha \{\tilde{s}'_{i_n}\}_{n=1}^N \mathbf{W}_{\text{res}}, \tag{7}$$

where $\{\tilde{z}_k^{(t)}\}_{k \in \{i_n\}_{n=1}^N}$ refers to the features of video tokens (at controlled keyframe positions) inside the model's DiT block, $\{\tilde{s}'_{i_n}\}_{n=1}^N$ refers to the features of sketch tokens inside the same DiT block, $\mathbf{W}_{\text{res}} \in \mathbb{R}^{D \times D}$ is a trainable weight, $D$ is the feature dimension of tokens.

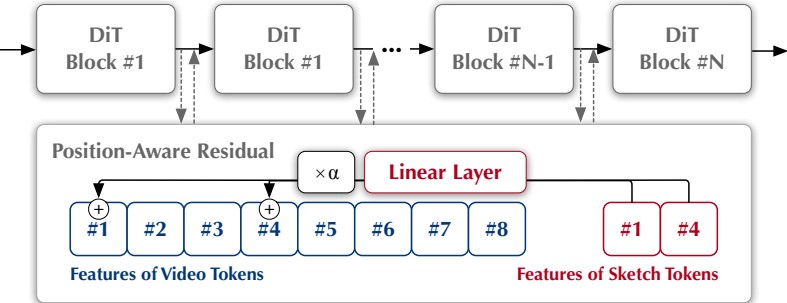

Figure 10: Illustration of the position-aware residual inside ToonComposer.

As shown in Figure 10, this operation is applied on output features of all DiT blocks except the last one. During training, the scale weight $\alpha$ is set to $1$. During inference, users can adjust the weight $\alpha$ to loosen or strengthen the control of keyframe sketches. The use case of the position-aware residual mechanism is illustrated in Section 4.4 of the main paper.

### A.2 DETAILS OF SLRA

The SLRA in ToonComposer is designed to effectively adapt video generation models to the cartoon domain while keeping its temporal video prior intact. In this section, we describe the inner process of SLRA step by step. Unlike conventional LoRA, the SLRA is designed to alter the spatial behaviors of the attention modules in a DiT model, so as to preserve its temporal prior, which is essential for video generation. The SLRA operates on a low-rank space and learns two matrices $\mathbf{W}^{\text{down}}$ and $\mathbf{W}^{\text{up}}$ to downsample the token feature to the low-rank space and to the original space. Let $H$ and $W$ be the spatial sizes of DiT's latent tokens, and $K$ and $N$ be the temporal length of the video tokens and sketch tokens. Given a token sequence $h \in \mathbb{R}^{L \times D}$ inside each self-attention module of $\epsilon_\theta$, where $L = (K + N) \times H \times W$ is the full length of the token sequence, SLRA operates first by downsampling the feature dimension of the input hidden token with a linear layer:

$$h^{\text{low}} = h \mathbf{W}^{\text{down}}, \tag{8}$$

where $\mathbf{W}^{\mathrm{down}} \in \mathbb{R}^{D \times D'}$ $D' \ll D$, yielding $h^{\mathrm{low}} \in \mathbb{R}^{L \times D'}$.

Then, SLRA reshapes $h^{\mathrm{low}}$ to $\tilde{h}^{\mathrm{low}} \in \mathbb{R}^{[K+N] \times [H \times W] \times D'}$, recovering their original spatial-temporal arrangements. After reshaping, we perform a self-attention operation on spatial dimension only. This is achieved by performing the attention mechanism on the spatial dimension of each frame **independently**:

$$Q = pos(\tilde{h}_l^{\mathrm{low}} \mathbf{W}_Q), \quad K = pos(\tilde{h}_l^{\mathrm{low}} \mathbf{W}_K), \quad V = \tilde{h}_l^{\mathrm{low}} \mathbf{W}_V \qquad (9)$$

$$O = \mathrm{softmax}\left(QK^T\right) V \mathbf{W}_O, \qquad (10)$$

where the subscript $l$ represents the index along the $l$-th temporal dimension, and $pos(\cdot)$ applies positional embeddings of corresponding spatial locations on queries and keys. Thus, the attention computation is performed within each frame. $\mathbf{W}_Q, \mathbf{W}_K, \mathbf{W}_V, \mathbf{W}_O \in \mathbb{R}^{D' \times D'}$ are trainable matrices, and $O \in \mathbb{R}^{H \times W \times D'}$. The same positional embeddings as the main model are applied to both video and sketch tokens during this attention operation. As a result, the propagation of information in the SLRA module is performed only in the spatial dimension, while leaving the temporal dimension intact.

Following that, we reshape $O$ to $\hat{h}^{\mathrm{low}} \in \mathbb{R}^{L \times D'}$ as a sequence, then upsample it to the original dimension:

$$h_{\mathrm{res}} = \hat{h}^{\mathrm{low}} \mathbf{W}^{\mathrm{up}}, \qquad (11)$$

where $\mathbf{W}^{\mathrm{up}} \in \mathbb{R}^{D' \times D}$. Finally, the adapted DiT model's self-attention output $\mathrm{SA}(h)$ is modified as:

$$h' = \mathrm{SA}(h) + h_{\mathrm{res}}. \qquad (12)$$

SLRA ensures that cartoon-specific spatial features are learned without disrupting temporal coherence, efficiently adapting a DiT-based video diffusion model to the cartoon domain.

## A.3 DETAILS OF THE THE SPARSE SKETCH INJECTION

In the main paper, we do not take the temporal compression of VAE into account for simplicity of notation. Here we show how we handle sparse sketches given that foundation model's VAE (Wan 2.1) compresses 4 temporal frames into a single latent frame as follows:

- *Encode with VAE.* We begin with sparse sketches in pixel space and construct a full-length sketch video by inserting all-zero (blank) sketch images at frames where no sketch is provided. We feed this sketch video through the same VAE encoder used for the RGB video (repeating the sketch across 3 RGB channels if necessary). This yields a sequence of latent frames, with one latent representing every 4 original frames (starting from the first frame). Maintaining the native encoding process of the Video VAE is crucial, as we found that directly encoding sketches frame-by-frame as images yields suboptimal results.

- *Discard Empty Latents.* Latent frames corresponding to 4 purely blank sketches are discarded. We preserve only those latents that contain at least one real sketch frame within their 4-frame block to serve as conditions for injection.

- *Add Binary Mask.* Simultaneously, we generate a binary temporal mask along the channel dimension to indicate which specific frame(s) within the 4-frame block of each preserved latent contain valid content. This mask follows the official Wan implementation for transforming temporal masks into latent space. The combination of the preserved latent (containing empty frames) and this binary mask ensures the model remains aware of the precise keyframe location within a 4-frame block (i.e., within a single VAE-encoded latent).

- *Sketch Projection.* We then employ an additional patch embedding linear layer (trained separately) to transform these latents into the internal dimension of the DiT tokens. These sketch tokens are appended to the end of the original token sequence of the DiT backbone.

- *Positional Encoding Mapping.* During the attention operations of the DiT, we assign these sketch tokens the RoPE temporal indices that match the video tokens at the corresponding temporal positions.

These steps ensure that the model correctly handles the VAE's temporal compression and maintains precise awareness of each sketch keyframe's temporal location.

# B  EXPERIMENTAL DETAILS

In this section, we present additional experimental results to complement the evaluations in the main paper, including the dataset processing pipeline, the human evaluation conducted on the real-world PKBench benchmark, qualitative comparisons showcasing ToonComposer's performance with human-drawn sketches.

## B.1  DATASET CONSTRUCTION PIPELINE

This section outlines the detailed pipeline for constructing the PKData dataset. The process involves four stages: video scene-cut segmentation, clip filtering, caption generation, and sketch generation. Each stage is carefully designed to ensure the dataset is diverse, high-quality, and suitable for training ToonComposer in the post-keyframing task.

**Video Scene-Cut Segmentation**    To create manageable and coherent video clips, we first segment the source cartoon videos into shorter clips using PySceneDetect [1], a scene detection library that identifies scene boundaries based on visual transitions. We get clips with an average duration of approximately 10 seconds after this process. This scene-cut segmentation ensures that each clip represents a single coherent scene, facilitating the generation of temporally consistent animations.

**Clip Filtering**    To ensure the quality and relevance of the dataset, we apply a rigorous filtering process to remove unsuitable clips. This process involves multiple criteria to eliminate low-quality or irrelevant content:

- **Rule-Based Filtering**: We discard clips containing all-white, all-black, or pure-color scenes, as these lack meaningful visual content for training.

- **Scene Consistency Check**: To ensure clips represent a single coherent scene, we compute the DINO (Caron et al., 2021) feature similarity between the first and last frames of each clip. Clips exhibiting drastic scene changes (i.e., low similarity scores) are removed to avoid abrupt transitions that could disrupt training.

- **Text Scene Detection**: Using Qwen-VL (Bai et al., 2023), a vision-language model, we identify and filter out clips dominated by pure text, which are irrelevant for animation generation.

- **Scene Cut Validation**: To verify the accuracy of scene-cut segmentation, we employ Qwen-VL (Bai et al., 2023) again to detect clips containing multiple scene cuts within a single clip. This is achieved by asking the VLM model if the first and last video frames are from the same scene. Clips with multiple scene cuts are removed from the training dataset.

Through this multi-step filtering, we reduce an initial pool of approximately 100K clips to 37K high-quality clips, ensuring each clip is visually coherent, content-rich, and suitable for training ToonComposer.

**Caption Generation**    To provide textual context for each clip, we generate descriptive captions using CogVLM (Wang et al., 2024a). The captions are designed to enhance the model's ability to understand and generate cartoon sequences by focusing on key visual and dynamic elements. We use the following prompt for caption generation: *"Describe this cartoon video in detail, ensuring that the main object(s) in the scene serves as the grammatical subject of your answer sentence. Focus on transformations throughout the video, including changes in color, lighting, and atmosphere, as well as the movements, actions, and interactions of characters or objects."* This prompt ensures that captions are detailed, action-oriented, and centered on the primary subjects, capturing essential transformations and interactions within each clip.

**Sketch Generation**    To enhance the robustness of ToonComposer across varied sketch styles in post-keyframing tasks, we generate diverse sketches for input video frames, reflecting the stylistic

---

[1] PySceneDetect is a scene cut detection and video splitting tool: https://www.scenedetect.com/.

variations arising from different artist preferences and tools. Specifically, we generate four distinct sketch versions per frame using open-source CNN-based models: two lineart models from ControlNet (Zhang & Agrawala, 2023), Anime2Sketch (Xiang et al., 2022), and Anyline (Soria et al., 2023). Additionally, we develop IC-Sketcher, an image-to-image generative model based on FLUX.1-dev [2], fine-tuned with in-context LoRA (Huang et al., 2024a) on a curated dataset of real human-drawn sketches from multiple artists. IC-Sketcher generates a fifth sketch variant that closely mimics real-world artistic styles, further enriching the dataset's diversity. These diverse sketches, illustrated in Figure 5, equips ToonComposer to handle a wide variety of sketch inputs, enhancing its applicability in practical cartoon production scenarios.

## B.2 HUMAN EVALUATION ON REAL BENCHMARK

To further investigate users' preferences on the generation results, we conducted human evaluations to compare the generation results produced by our method and other baselines. We randomly select 30 samples from the benchmarks and generate cartoon videos for each method using the aforementioned pipeline. Our evaluation process involved 47 participants, each of whom was asked to select the video with the best aesthetic quality and motion quality. The results are shown in Table 5, where our method achieves the highest win rate on both metrics, significantly exceeding the second best competitor.

Table 5: User preference rates for aesthetic quality (A. Q.) and motion quality (M. Q.) of cartoons generated by ToonCrafter (Xing et al., 2024a), AniDoc (Meng et al., 2024), LVCD (Huang et al., 2024b), and our ToonComposer.

| Method | Aesthetic Quality↑ | Motion Quality↑ |
|---|---|---|
| AniDoc | 4.45% | 5.34% |
| LVCD | 7.54% | 7.91% |
| ToonCrafter | 17.02% | 18.19% |
| ToonComposer (Ours) | **70.99%** | **68.58%** |

## B.3 QUALITATIVE COMPARISON ON REAL BENCHMARK

Figure 11 visualizes the comparison between all methods, with zoom-in patches from a randomly selected region provided in the rightmost column. It is observed that previous methods deviate from the overall style of the first reference frame. Specifically, ToonCrafter generates intermediate frames with prominent bold lines, likely influenced by the bold brush strokes in the human-drawn sketches, revealing its limited robustness to diverse sketch styles. In contrast, our ToonComposer produces video frames with superior visual quality, motion coherence, and style consistency, consistent with the quantitative results.

## C MORE ANALYSIS

This section provides more comprehensive analyses of ToonComposer's performance and design choices. Specifically, we discuss 1) the model's controllability with varying numbers of keyframe sketches, 2) compare its performance against baselines with increased sketch inputs, 3) ablate key architectural components (sparse sketch injection, positional-encoding mapping, and position-aware residual), 4) evaluate SLRA against LoRA with different ranks, 5) assess the impact of text prompts, 6) discuss its generalization to 3D animation, and 7) discuss the limitation of current model.

### C.1 COMPARISON USING MULTIPLE SKETCHES

In the primary experiments, all models were evaluated using two keyframe sketches. To assess performance with additional inputs, we extended the evaluation to four sketches for each model. As reported in Table 6, ToonComposer outperforms all baselines when limited to two sketches, which even surpasses the results of all baseline methods with four sketches. This outcome underscores

---

[2] FLUX.1-dev image generation model: `https://github.com/black-forest-labs/flux`.

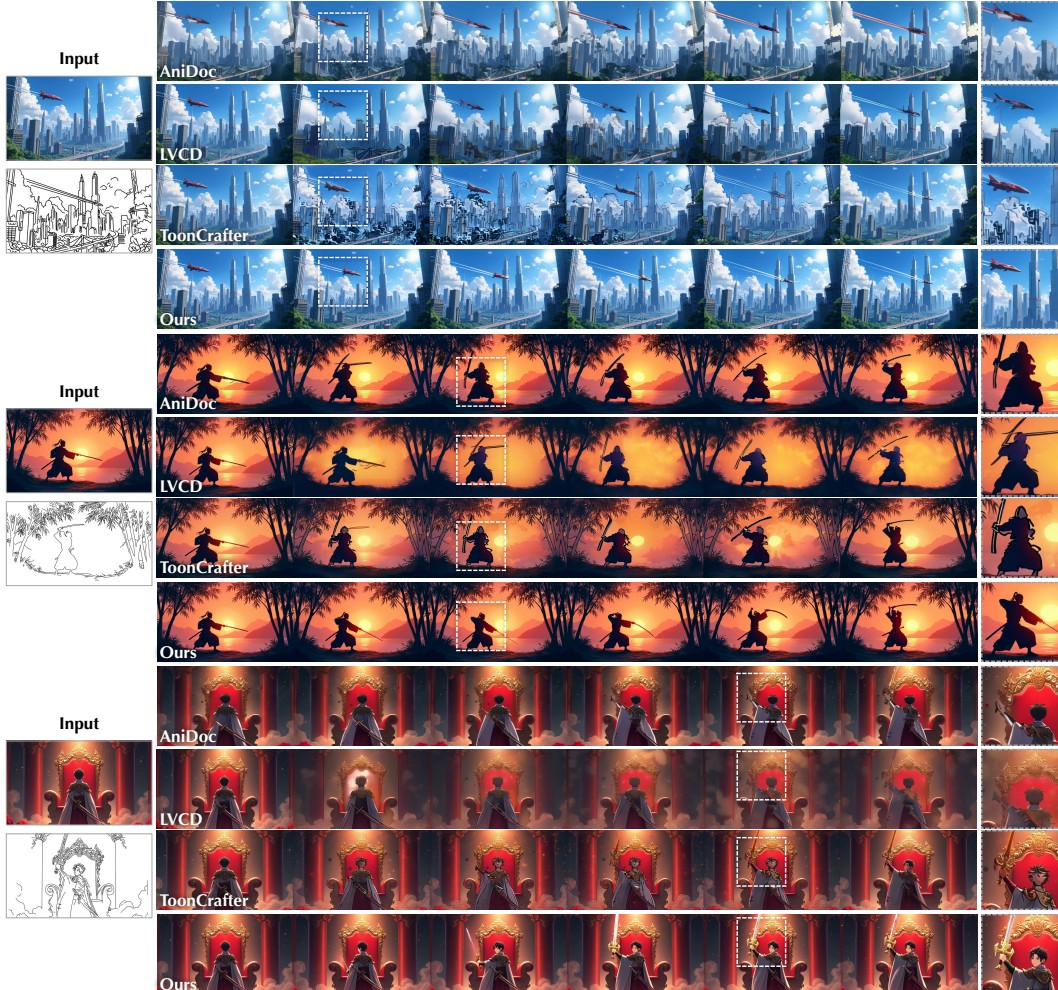

Figure 11: Comparison on the benchmark PKBench, using keyframe sketches drawn by the human artists. Zoom-in patches are shown in the rightmost column. Our method generates high-quality results from real sketch inputs, whereas other methods struggle to maintain visual consistency. Please refer to the supplementary video for additional examples.

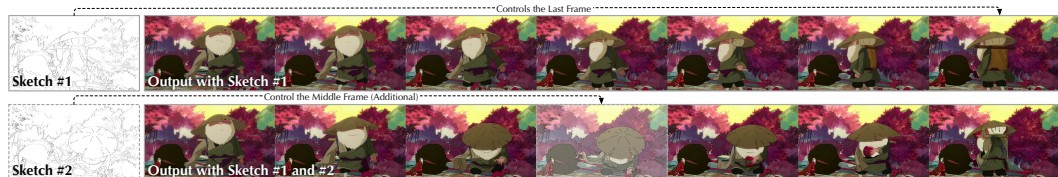

Figure 12: ToonComposer's flexible controllability with varying keyframe sketches. Using only sketch #1 as the final keyframe and the prompt "an old man turns back," ToonComposer generates a sequence where the old man turns directly (first row). Adding sketch #2 to control the middle keyframe, while keeping the prompt unchanged, results in a sequence where the old man first picks up a fruit before turning back (second row).

ToonComposer's robustness and efficiency, while also exposing the error accumulation challenges inherent in prior two-stage approaches.

Table 6: Performance comparison with varying numbers of sketches on the synthetic benchmark.

| Model | Sketch Num | LPIPS↓ | DISTS↓ | CLIP↑ |
|---|---|---|---|---|
| ToonComposer | 2 | 0.1785 | 0.0926 | 0.9449 |
| | 4 | **0.0882** | **0.0656** | **0.9636** |
| ToonCrafter | 2 | 0.3830 | 0.5571 | 0.8463 |
| | 4 | 0.2956 | 0.5247 | 0.8973 |
| LVCD | 2 | 0.3910 | 0.5505 | 0.8428 |
| | 4 | 0.3711 | 0.5558 | 0.8495 |
| AniDoc | 2 | 0.3734 | 0.5461 | 0.8665 |
| | 4 | 0.3076 | 0.5378 | 0.8880 |

Table 7: Comparison of SLRA and LoRA adaptations with varying ranks (value of $D'$ in SLRA).

| Rank | Adaptation | Parameter Count | LPIPS↓ | DISTS↓ | CLIP↑ |
|---|---|---|---|---|---|
| 24 | SLRA | 37M | **0.1742** | 0.0975 | **0.9642** |
| | LoRA | 105M | 0.1922 | 0.1082 | 0.9628 |
| 144 | SLRA | 89M | 0.1874 | **0.0955** | 0.9634 |
| | LoRA | 499M | 0.2076 | 0.1218 | 0.9586 |
| 256 | SLRA | 134M | 0.1785 | 0.1046 | 0.9597 |
| | LoRA | 866M | 0.2114 | 0.1342 | 0.9591 |

## C.2 CONTROLLABILITY WITH INCREASING KEYFRAME SKETCHES

The sparse sketch injection mechanism of ToonComposer enables flexible control by supporting a variable number of input keyframe sketches, increasing its utility in the cartoon production pipeline. This adaptability allows artists to balance creative control and automation based on the complexity of the desired motion. As shown in Figure 12, we demonstrate the ability of ToonComposer to generate distinct cartoon sequences from different numbers of input sketches, all conditioned on the same text prompt. Additional examples are available in the supplementary video, which illustrates the versatility of our method in diverse scenarios.

## C.3 COMPARISON BETWEEN SLRA AND LoRA WITH VARYING RANKS

To further validate SLRA's efficiency and effectiveness, we compare SLRA models with LoRA models with varying ranks of 24, 144, and 256. Rank here refers to the value of $D'$ described in Section 3.3. As shown in Table 7, all SLRA models outperform their LoRA counterparts, even with significantly fewer trainable parameters with the same rank (indicated by Parameter Count). This highlights SLRA's superiority in adapting the model while preserving temporal consistency. We empirically observe that larger ranks do not guarantee better results.

Table 8: Performance comparison with and without text prompts on the synthetic benchmark.

| Model | Prompt | LPIPS↓ | DISTS↓ | CLIP↑ |
|---|---|---|---|---|
| ToonComposer | w/ | **0.1785** | **0.0926** | 0.9449 |
| ToonComposer | w/o | 0.2091 | 0.0941 | **0.9517** |
| ToonCrafter | w/ | 0.3830 | 0.5571 | 0.8463 |
| LVCD | w/o | 0.3910 | 0.5505 | 0.8428 |
| AniDoc | w/o | 0.3734 | 0.5461 | 0.8665 |

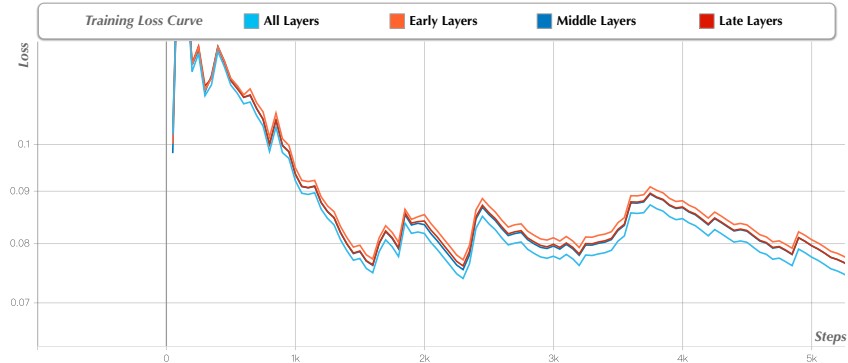

Figure 13: The training loss curve of four different choice of DiT layers (blocks) to apply SLRA: all layers, early (1/3) layers, middle (1/3) layers, and late (1/3) layers. All other training settings and random states are all controlled. The model with all layers applied SLRA shows lowest loss, while the models with middle and late layers applied with SLRA shows comprable losses that are slightly higher than the all-layer variant. The model with only early layers applied with SLRA reports highest loss value, indicating that early layers contribute less to the cartoon domain adaptation compared to other layers.

## C.4 ANALYSIS ON DIFFERENT LAYER PLACEMENT OF SLRA

To explore which part of layers (blocks) inside of the DiT that contribute more in the adaptation of SLRA, we train four variants of ToonComposer with controlled training settings and random states to compare their performance. Specifically, we train four models where all layers, early (1/3) layers, middle (1/3) layers, and late (1/3) layers, are applied with SLRA. The training loss curves are shown in Figure 13 and the numerical test results are shown in Table 9. The model with all layers applied SLRA shows lowest loss, while the models with middle and late layers applied with SLRA shows comprable losses that are slightly higher than the all-layer variant. The model with only early layers applied with SLRA reports highest loss value, indicating that early layers contribute less to the cartoon domain adaptation compared to other layers. Both the training loss and the test results indicate that the all-layer placement of SLRA is the most suitable choice in our design.

Table 9: Numerical results of models with different placement of SLRA.

| SLRA Placement | LPIPS↓ | DISTS↓ | CLIP↑ |
|---|---|---|---|
| Early Layers | 0.1810 | 0.1165 | 0.9379 |
| Middle Layers | 0.1857 | 0.1194 | 0.9386 |
| Late Layers | 0.1809 | 0.1174 | 0.9367 |
| All Layers | **0.1765** | **0.1138** | **0.9392** |

## C.5 EFFECT OF TEXT PROMPTS

In video generation tasks, text prompts supply contextual details about the scene, enabling video generation models to produce outputs that better align with user intentions. In our experiments, the prior cartoon inbetweening model ToonCrafter (Xing et al., 2024a) requires prompts, whereas the colorization models LVCD (Huang et al., 2024b) and AniDoc (Meng et al., 2024) lack support for them. In ToonComposer, text prompts are treated as optional inputs in the post-keyframing process, primarily serving to resolve ambiguities.

To evaluate ToonComposer's robustness in the absence of prompts, we test it on the synthetic benchmark using empty prompts. As reported in Table 8, our model continues to outperform all baselines even without prompts, demonstrating its strong performance even without textual guidance.

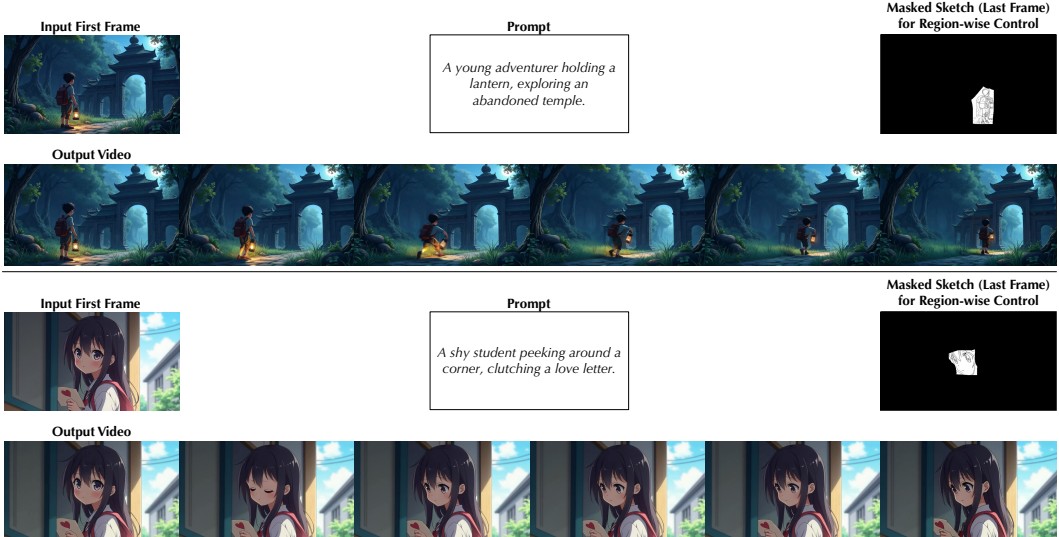

Figure 14: More use samples of region-wise control. Our model performs well when the sketch mask is extremely large. The model generate faithful content to the given sketch part and produce plausible and coherent content in masked areas.

## C.6    GENERALIZATION TO 3D ANIMATION

Although 3D animation production pipelines differ from 2D cartoon, ToonComposer can be extended to 3D-rendered animations by adapting the initial reference frame to a 3D-rendered image. We finetune the model on a small dataset of 3D animation clips, allowing it to generate high-quality sequences in a 3D style while adhering to the post-keyframing paradigm. This extension underscores ToonComposer's versatility and its potential for applications beyond traditional 2D cartoons. Samples of these 3D animations are included in the ***supplementary video***.

## C.7    DETAILS OF REGION-WISE CONTROL

**Discussion of the Training Strategy.**    During training, we draw 3 types of connected geometric shapes (rectangles, circles, polygons) with random locations, sizes, aspect ratios, and coverage up to 60% of the frame. This encourages the model to fill in missing contiguous regions, rather than dealing with isolated pixels. For the training strategy, we chose random masks over semantic masks for two key reasons. The first reason is alignment with user behavior. At inference time, users typically provide coarse masks that cover general areas rather than single precise object segmentations. Geometric masks simulate this "rough erasure" more effectively than perfect semantic masks. The second reason is generalizability. Training with arbitrary geometric shapes encourages the model to develop a more robust contextual reasoning ability. It learns to generate plausible content based on surrounding textures and text prompts, regardless of the semantic category. In contrast, training solely on semantic masks might restrict the model to only filling specific, recognized object classes, reducing robustness to user input.

**More Results.**    We provide more results of region-wise control with large masks in addition to those presented in Figure 7. As shown in Figure 14, when users leave a large region blank and mark it as to be filled, the model uses surrounding layout, motion, and the text prompt to generate plausible content in that region. This indicates the robustness of our region-wise control training strategy.

## C.8    SKETCH FIDELITY EVALUATION

We have provided visual overlay samples in our supplementary video to show the sketch faithfulness qualitatively, we further conduct a sketch–frame alignment evaluation quantitatively. For each

method, we run its output through the same sketch extractor used for creating synthetic sketches. At each keyframe index, we compare the extracted sketch with the input sketch using SSIM and the binarized pixel-level accuracy and report these numbers for all methods in Table 10. This complements the existing metrics and directly measures adherence to the given keyframe sketch input. ToonComposer achieves significantly higher sketch fidelity compared to baselines, confirming its precise temporal alignment.

Table 10: Quantitative results of the sketch fidelity of each model.

| Model | Sketch SSIM↑ | Sketch Binary Accuracy↑ |
| --- | --- | --- |
| AniDoc | 0.5047 | 0.8196 |
| LVCD | 0.5265 | 0.8347 |
| ToonCrafter | 0.5278 | 0.8327 |
| ToonComposer | **0.8360** | **0.9444** |

## C.9 LIMITATIONS

Although employing data augmentation to enhance robustness, ToonComposer may experience performance degradation when handling drastic sketch style variations, such as thick ink strokes, which can lead to reduced alignment with keyframe details in highly unconventional cases. Moreover, the computational cost of the underlying video diffusion model remains substantial, requiring significant resources for training and inference, which may limit accessibility in resource-constrained settings. Optimizing the model's inference speed presents a valuable direction for future research.

## D SUPPLEMENTARY VIDEO

To provide a comprehensive demonstration of ToonComposer's capabilities, the authors include a video in the submission's supplementary materials (in the ZIP file). This video showcases qualitative results, including generated cartoon sequences, comparisons with baseline methods, and examples of region-wise control, multiple-keyframe controllability, and generalization to 3D animation. The authors encourage readers to refer to the supplementary video to evaluate the visual quality, motion coherence, and versatility of ToonComposer, as dynamic video content more effectively conveys these attributes than static images in the manuscript.

## E USE OF LARGE LANGUAGE MODELS

The authors utilized large language models only for minor polishing of the manuscript's writing. LLMs did not contribute to research ideation or experimental design. The authors take full responsibility for all contents of the paper.

