# OpenReview forum: "ToonComposer: Streamlining Cartoon Production with Generative Post-Keyframing"
_ICLR.cc/2026/Conference — ICLR 2026 Poster_

### Official Review · Reviewer_HHzB · 2025-10-26

**Soundness:** 4
**Presentation:** 4
**Contribution:** 4
**Rating:** 8
**Confidence:** 4

**Summary:**

The paper proposes ToonComposer, a diffusion-transformer (DiT)–based framework targeting the post-keyframing stage of cartoon production. It unifies inbetweening and colorization into a single stage to better handle large motions between keyframes. To adapt a modern video foundation model to the cartoon domain while preserving temporal priors, the authors introduce a Spatial Low-Rank Adapter (SLRA) that constrains adaptation primarily in the spatial pathway. The paper also present PKBench, a benchmark of human-drawn keyframe sketches and corresponding reference color frames intended to reflect real production workflows. On PKData, ToonComposer reportedly outperforms baselines on LPIPS↓, DISTS↓, and a CLIP-based similarity metric.

**Strengths:**

1. Addressing the post-keyframing stages (unifying inbetweening and colorization) is highly relevant to real animation pipelines, especially under large motion. The motivation is strong.
2. PKBench Benchmark is a useful step and notiable contribution toward standardized evaluation, with multiple sketch styles per clip and paired references that resemble real use cases.
3. Consistent improvements on LPIPS/DISTS/CLIP (on PKData) indicates the approach is competitive with existing baselines. Both qualitative and quantitative results are good.

**Weaknesses:**

1. The only concern is single-dataset evaluation. Results are reported only on PKData, which limits claims about generalization (e.g., to different studios, styles, or line quality). But if PKData is sufficiently large and diverse, please document that to support the claim.

2. Data diversity clarity. The breadth and distribution of PKData is important for evaluation, please specify. Provide counts/percentages per category and a few representative examples.

3. Sketch provenance & difficulty. Parts of the paper suggest sketches may be derived with tools such as ControlNet/Anime2Sketch/AnyLine; elsewhere they are described as human-drawn. The exact provenance and the proportion of truly freehand, rough sketches vs. algorithmically derived outlines need clarification. If most inputs are algorithmic edge/sketch maps, robustness to messy, expressive, or incomplete human sketches remains uncertain. Does the model leave any capiblity for such test case?

4. Temporal-prior preservation evidence. While SLRA is motivated to “preserve temporal priors,” the empirical evidence is thin. There’s no direct analysis showing temporal coherence is maintained (or improved) relative to alternatives. A granular ablation would strengthen the methodological claims.

**Questions:**

1. PKData diversity. What is the diversity of PKData, could you list some examples or categories numbers?

2. Sketch provenance. What fraction of PKBench sketches are genuinely human-drawn freehand vs. automatically extracted (e.g., ControlNet/Anime2Sketch/AnyLine)? Briefly describe the differences you observe between the two.

---

> ### Author Response · Authors · 2025-11-28
> **Response to Reviewer HHzB**
>
> ### 1. **Details of the Dataset**
>
> PKData comprises 37K high-quality cartoon/anime video clips, selected through an aggressive filtering process focusing on aesthetic and motion quality (detailed in Appendix B). The dataset spans a broad spectrum of cartoon themes and compositions, including but not limited to urban, natural, fantasy, indoor scenes, and close-up character shots. Regarding the specific categories, as the data is sourced from licensed and internal materials, we are unable to release detailed statistics. However, we emphasize that the dataset's internal diversity is substantial enough to train a robust foundation. Furthermore, the model's successful generalization to the human-drawn PKBench serves as strong empirical evidence that PKData covers the necessary variance to support our claims. The human-drawn PKBench will be released for community use.
>
> ### 2. **Sketch provenance and difficulty**
>
> For training on PKData, we use multiple types of algorithmic sketches to enhance the diversity, including four variants produced by open-source lineart/sketch models (two ControlNet lineart models, Anime2Sketch, AnyLine), plus the fifth variant generated by IC-Sketcher (a FLUX.1-dev based model trained with in-context LoRA on a small set of human sketches). These automatically generated sketches differ in line thickness and abstraction.
>
> For PKBench, **all sketches** are human-drawn on drawing tablets by professional artists, which are used in our experiments for real scenario evaluation and the user study. These human-drawn sketches are rougher and sometimes incomplete. Figure 5 in the main paper illustrates the visual difference.
>
> Despite this gap between synthetic training data and real-world testing data, ToonComposer demonstrates reasonable generalization. It achieves superior performance on PKBench across quantitative metrics (Table 2), qualitative visualizations (Appendix Fig. 10), and user preference studies (Appendix Table 5). We have updated the manuscript to explicitly frame this as an evaluation of robustness to out-of-distribution, real-world inputs.
>
> ### 3. **Temporal-prior preservation evidence**
>
> Thanks for the insightful suggestion. Our motivation for SLRA is to adapt the appearance to a cartoon style while preserving the temporal prior of the base video DiT. We already reported the ablation results in Table 2 in the main paper for SLRA, which indicate our SLRA performs better than competitor like LoRA in overall visual quality when trained under the same settings. To further illustrate the temporal coherence, we use the VBench motion smoothness metrics and Fréchet Video Distance (FVD) to compare the SLRA to LoRA, as shown in the table below.
>
> ------
> | Model | Motion Consistency↑ | FVD↓ |
> | :------:| :------:|:------:|
> | LoRA  | 0.9884 | 47.2428 |
> | SLRA (Ours) | **0.9906** | **46.8062** |
> ------
>
> The reported results make it clearer that SLRA maintains the temporal quality compared to conventional LoRA while improving general visual quality in the cartoon domain.

---

### Official Review · Reviewer_ehPw · 2025-10-28

**Soundness:** 3
**Presentation:** 3
**Contribution:** 1
**Rating:** 2
**Confidence:** 5

**Summary:**

ToonComposer is a one-stage, sketch-conditioned video diffusion system that unifies inbetweening and colorization for cartoons. It adds (i) sparse sketch injection with a position-aware residual so a few keyframe sketches can control layout/style over long clips, and (ii) a Spatial Low-Rank Adapter (SLRA) inside DiT attention that adapts appearance per-frame (spatial-only) while preserving temporal priors. Trained on a new PKData pipeline and evaluated with synthetic metrics and PKBench, it reports stronger perceptual quality and motion/style consistency than AniDoc, LVCD, and ToonCrafter.

**Strengths:**

* Introduces a single-stage post-keyframing formulation that unifies inbetweening and colorization—reducing the error accumulation inherent to two-stage pipelines and enabling control from as little as one sketch + one colored frame.

* Proposes sparse sketch injection with position-aware residual and positional-encoding mapping (sketch tokens appended to the latent token sequence with remapped RoPE to target specific frame indices); this is a clean, DiT-native control mechanism distinct from channel-wise concat.

* On a synthetic benchmark, the method scales control with more sketches and maintains strong perceptual scores, highlighting practical utility for real pipelines.

**Weaknesses:**

* **Using DiT is not novel:** Just changing unet-based diffusion with a DiT-based model is not a significant gain, and I can't see any specification. I don't know why the author bolded it. Lines 089-099.

* **SLRA is not novel:** Using a low-rank adapter for temporal or spatial adaptation is not novel, and many studies have used it for domain adaptation.

* **Sketch faithfulness is under-measured:** Current metrics (LPIPS/DISTS/CLIP) correlate with perceptual quality but not how well frames obey the sparse sketches. Add explicit sketch-to-frame alignment metrics.

* **Grounding/attribution gap:** The method claims precise control via position-aware residuals, but there’s no quantitative isolation of which regions are controlled by which sketch tokens.

* **Generality and portability unclear:** Training is tied to a specific high-capacity base model. Demonstrate portability by fine-tuning an open AnimateDiff [1] with SLRA and reporting deltas. Also show results on non-cartoon domains (simple line-art or cel-style outside the training distribution) to define failure boundaries.


* **Baselines miss sketch-specific controls:**  Comparisons emphasize two-stage cartoon baselines; fewer controls vs. modern controllable video models (e.g., ControlNet-style adapters on AnimateDiff [1], token-tracking/warping methods, or sketch-to-image extended temporally). Re-implement a sketch-ControlNet (edge/sketch map as conditioning) on a strong open video base and a two-stage ToonCrafter+colorization tuned for this data to ensure apples-to-apples.



[1] Guo, Yuwei, et al. "Animatediff: Animate your personalized text-to-image diffusion models without specific tuning." arXiv preprint arXiv:2307.04725 (2023).

**Questions:**

* **SLRA design choices:** SLRA applies per-frame spatial attention with shared positional embeddings. Have you tested where to place SLRA (early/mid/late layers) and rank-vs-quality-vs-latency trade-offs? A sweep would clarify efficiency claims and when a higher rank stops helping.

* **Region-wise control reliability:** Figure 7 is compelling, but how robust is region-wise control when sketches leave large areas blank or contain contradictory cues across keyframes?


**I am open to changing my score based on the author's responses.**

---

> ### Author Response · Authors · 2025-11-28
> **Response to Reviewer ehPw (Part 1)**
>
> ### 1. **Novelty**
>
> Thanks for your questions. We agree that the single point of replacing a UNet with a DiT or solely using a low-rank adapter are not novel in themselves. As shown in our paper title, we respectfully clarify that the contributions we intend to highlight are:
>
>   * **The post-keyframing formulation**, which directly generates fully colored cartoon sequences from very sparse keyframe sketches and one colored frame, removing the need for a separate inbetweening stage followed by a separate colorization stage. This design is motivated by how studio animators actually work.
>
>   * **The pipeline and techniques inside that effectively turn post-keyframing into practice**, where sketches are integrated as tokens via sparse sketch injection with positional encoding mapping, and a spatial cartoon domain adaptation with SLRA tailored to spatio-temporal-coupled modules.
>
> Since our model is not based on a spatial/temporal-decoupled foundation model (where we could literally tune the corresponding layers). SLRA is our way to recover the ability to change appearance while leaving temporal behavior governed by the pretrained prior.
>
> As shown in Table 3 of the main paper and the corresponding ablations, SLRA consistently outperforms LoRA and other variants under comparable parameter counts. We have revised our introduction to clarify our contribution respectively.
>
>
> ### 2. **Sketch faithfulness and control attribution**
>
> Thanks for the insightful suggestions. Although we have provided visual overlay samples in our supplementary video to show the sketch faithfulness qualitatively, we agree that the current metrics in the paper may not be straightforward to measure how well the generated frames follow the sparse sketches quantitatively.
>
> In response, the revised version adds a sketch–frame alignment evaluation at keyframe indices. For each method, we run its output through the same sketch extractor used for creating synthetic sketches. At each keyframe index, we compare the extracted sketch to the input sketch using SSIM and the binarized pixel-level accuracy and report these numbers for all methods in a new table (shown below).
>
> ------
> | Model | Sketch SSIM↑ | Sketch Binary Accuracy↑ |
> |:-------:|:------:|:-----:|
> | AniDoc | 0.5047 | 0.8196 |
> | LVCD | 0.5265 | 0.8347  |
> | ToonCrafter | 0.5278 | 0.8327 |
> | ToonComposer | **0.8360** | **0.9444** |
> ------
>
> This complements the existing metrics and directly measures adherence to the given keyframe sketch input. ToonComposer achieves significantly higher sketch fidelity compared to baselines, confirming precise temporal alignment. This is also reflected in Appendix C.8 of the revised paper.
>
> For the position-aware residual mentioned by the reviewer, we would like to clarify that the position-aware residual is an optional global strength control at inference time, where users can loosen the sketch control to produce diverse results. We have included a detailed discussion of the position-aware residual usage sample and ablation in the main paper.
>
>
> ### 3. **Portability**
>
> To address portability, we also apply the same design (sparse sketch injection + SLRA) to a smaller Wan 1.3B model and include its results in the revised paper. Results are shown in the table below.
>
> ------
> |Method|LPIPS↓|DISTS↓|CLIP↑|Subject Con.↑|Motion Smo.↑|Background Con.↑|Aesthetic Qua.↑|
> |:-:|:-:|:-:|:-:|:-:|:-:|:-:|:-:|
> |AniDoc|0.3734|0.5461|0.8665|0.9067|0.9798|0.9408|0.4962|
> |LVCD|0.3910|0.5505|0.8428|0.8316|0.9810|0.9183|0.4984|
> |ToonCrafter|0.3830|0.5571|0.8463|0.8075|0.9550|0.8920|0.5035|
> |ToonComposer (1.3B)|**0.1698**|0.1097|0.9292|0.9243|**0.9889**|0.9505|0.5576|
> |ToonComposer (14B)|0.1785|**0.0926**|**0.9449**|**0.9451**|0.9886|**0.9547**|**0.5999**|
> ------
>
> Although the absolute quality is slightly lower due to model capacity, the relative improvements of our ToonComposer remain consistent. This indicates that the method is **not tied** to a single large (14B) foundation model.
>
> For the adaptation to other modality, we also tried training ToonComposer on a set of rendered 3D animation, as samples included in our demo video, which also indicates the versatility of our method in scope, in addition to anime/cartoon. We would also like to propose the adaptation to more domain as an interesting future work.

---

> ### Author Response · Authors · 2025-11-28
> **Response to Reviewer ehPw (Part 2)**
>
> ### 4. **Comparison with CtrlNet-like Architecture**
>
> In addition, upon the request of the reviewer, we also train a new CtrlNet-based model on the AnimateDiff.
>
> Since the AnimateDiff is relatively outdated and to improve the fairness, we also train another CtrlNet variant based on the same foundation model (Wan 2.1 14B) of ToonComposer in the identical settings of our ablation experiment (Table 3 of the main paper).
>
> Specifically, the CtrlNet-based AnimateDiff version adopts the conventional CtrlNet manner, while the CtrlNet-based Wan version uses an extra trainable DiT block that shares the structure with the original DiT blocks as the CtrlNet module. The output of this CtrlNet block is added to the intermediate output of the original DiT block, except for the last one. We also train this added CtrlNet block along with LoRAs added to the DiT backbone, because we found that solely training the added CtrlNet block on the DiT model does not converge. Results are shown in the table below.
>
>
> ------
> |Method|LPIPS↓|DISTS↓|CLIP↑|
> |:-:|:-:|:-:|:-:|
> |AnimateDiff + CtrlNet |  0.3204  |  0.4922  |  0.8726  |
> |Wan2.1 + CtrlNet | 0.1962 | 0.1287 | 0.9155 |
> |Wan2.1 + Sparse Sketch Injection + LoRA|0.1922 | 0.1082 | 0.9628 |
> |Wan2.1 + Sparse Sketch Injection + SLRA (Ours)| **0.1874** | **0.0955** | **0.9634** |
> ------
>
> We can clearly see that our model outperforms all other CtrlNet-based variants, including the model based on AnimateDiff or the model based on the large-scale foundation model Wan 2.1 14B.
>
>
> ### 5. **SLRA design choices: placement and rank**
>
> We place SLRA inside every self-attention block of the DiT by default.
>
> As suggested by the reviewer, we rerun the training of models by applying SLRA only to early, middle, or late layers. Specifically, we trained four variants of ToonComposer where different DiT parts of layers (blocks) are applied with SLRA: all layers, early 1/3 layers, middle 1/3 layers, and late 1/3 layers. All random states and training settings other than the SLRA placement are all controlled. Due to the time limit, we train all models for 6000 steps for a fair comparison.
>
> We haved included training loss curves of these variant in Appendix C.4 of our revised paper. The model with all layers applied with SLRA shows the lowest loss. In addition, we report the test metrics in the table below.
>
> ------
> | SLRA Placement | LPIPS ↓ | DISTS ↓ | CLIP ↑  |
> |:-:|:-:|:-:|:--:|
> |Early 1/3 Layers| 0.1810|0.1165|0.9379|
> |Middle 1/3 Layers| 0.1857|0.1194|0.9386|
> |Late 1/3 Layers| 0.1809|0.1174|0.9367|
> |All Layers| **0.1765**|**0.1138**|**0.9392**|
> ------
>
> These results indicate that the all-layer placement of SLRA is the most suitable choice in our design, as the differences in parameter count are minor.
>
> For the choice of ranks for SLRA, we already include an analysis in Appendix C.3 (Table "Comparison of SLRA and LoRA adaptations with varying ranks"), where we vary the internal dimension $D'$ and compare against LoRA with comparable parameter counts, and the model with rank of 144 reports the overall best performance.
>
> ### 6. **Region-wise control robustness**
>
> When users leave a large region blank and mark it as to be filled, the model uses the surrounding layout, motion, and the text prompt to generate plausible content in that region. More visual samples are included in Figure 13 of Appendix C.7.
>
> In addition, if sketches across keyframes give conflicting cues about a masked region (e.g., completely unrelated content from different scenecuts), these inputs fall out of our post-keyframing task scope. We provide additional results and a discussion of the region-wise control in Appendix C.7 of the revised paper.

---

### Official Review · Reviewer_pGDP · 2025-10-30

**Soundness:** 3
**Presentation:** 2
**Contribution:** 3
**Rating:** 6
**Confidence:** 4

**Summary:**

This paper introduces ToonComposer, a novel generative model designed to streamline cartoon and anime production by unifying the inbetweening (interpolation) and colorization stages into a single "post-keyframing" process. Traditional animation workflows often involve these stages separately, leading to intensive manual effort and potential for error accumulation. By taking sparse sketches and color references as input, ToonComposer aims to automate this critical part of the production pipeline.

**Strengths:**

- The core contribution is the unification of sketch colorization and interpolation into a single, cohesive task. This approach has the potential to significantly streamline the anime production workflow.

- The curation of the large-scale PKData dataset and the development of the PKBench benchmark, which uniquely includes human-drawn sketches, are valuable resources. They facilitate robust training and a more rigorous evaluation of cartoon generation models than previously possible.

- The paper provides strong experimental results, showing both quantitative and qualitative improvements in animation video generation. The ablation studies are thorough and effectively demonstrate the contribution of key components, such as the SLRA.

**Weaknesses:**

- The paper uses sparse sketches and color references as simultaneous control conditions. However, the concept of using multiple controls is not entirely novel and has been explored in prior works (e.g., LayerAnimate[1]). The authors should more explicitly discuss the unique advantages of their specific approach in the animation workflow.

- Temporal Alignment: The VAE in Wan employs a 4x temporal compression, meaning each latent token encapsulates information from multiple frames. The paper states that temporal position embeddings (using RoPE with temporal index $j$) and a "position-aware residual" are used to enforce control at specific keyframe indices. However, these indices ($j$) operate in the compressed latent space. It is unclear how ToonComposer guarantees that the decoded video aligns precisely with the input sketches at the correct corresponding pixel-space frame number. The paper lacks a detailed discussion or, more importantly, a visual analysis (e.g., frame-by-frame comparison at keyframes) to validate this crucial temporal alignment.

- Minor Issues:
    - The paper does not specify the implementation for the "sketch projection."
    - The ablation study in Table 6 clearly indicates that the Position-Aware Residual is a critical component for ToonComposer. Given its importance, this component and its analysis should be moved from the appendix to the main body of the paper.
    - There is an extra parenthesis ')' on page 5, line 234.

[1] LayerAnimate: Layer-level Control for Animation. Yuxue Yang, Lue Fan, Zuzeng Lin, Feng Wang, Zhaoxiang Zhang; Proceedings of the IEEE/CVF International Conference on Computer Vision (ICCV), 2025, pp. 10865-10874

**Questions:**

In Section 3.3 (Region-Wise Control), the model is trained using random masks to remove sketch regions. The description suggests these might be pixel-wise random masks. If this is the case, the training masks would resemble "noise", which is fundamentally different from the semantically coherent masks (e.g., masking the entire "train" in Figure 7) used during inference.

Does this significant discrepancy between the distribution of training masks and inference masks affect the model's practical performance? Why was this training strategy chosen over using more semantic masks during training?

---

> ### Author Response · Authors · 2025-11-28
> **Response to Reviewer pGDP (Part 1)**
>
> ### 1. **Contribution**
>
> We would like to appreciate the reviewer pGDP for recognizing the value of our post-keyframing paradigm that unifies colorization and interpolation in the strength part.
>
> For the question about our contribution, we would like to emphasize that while multiple controls exist in interesting related works like LayerAnimate, our contribution from the task aspect is the **post-keyframing** paradigm.
>
> In addition to that, our contribution lies in how these controls are used and in the task formulation of the post-keyframing. ToonComposer is designed around the post-keyframing stage. It directly generates fully colored cartoon sequences from very sparse control, typically one colored frame plus one or a few keyframe sketches, and avoids a separate inbetweening-then-colorization pipeline.
>
> This design is motivated by how studio animators actually work: keyframes are created by senior artists, and the remaining frames and colorization are tedious and repetitive.
>
> Although the great related work LayerAnimate supports multiple controls, it adopts per-frame sketches as input control, while our ToonComposer assumes extremely sparse skethces. We have clarified this positioning in the introduction and related work.
>
>
>
> ### 2. **Temporal alignment under compression and the sketch projection**
>
> We thank the reviewer for the question regarding the temporal alignment results of our method.
>
> **We would like to remind that the keyframe alignment visualization is provided in the supplmentary video, and we encourage the reviewers to view it.**
>
>
> In addition, we show how we handle sparse sketches given that Wan's VAE compresses 4 frames into a single latent frame as follows:
>   *  **Step 1. Encode with VAE.** We begin with sparse sketches in pixel space and construct a full-length sketch video by inserting all-zero (blank) sketch images at frames where no sketch is provided. We feed this sketch video through the same VAE encoder used for the RGB video (repeating the sketch across 3 RGB channels if necessary). This yields a sequence of latent frames, with one latent representing every 4 original frames (starting from the first frame). Maintaining the native encoding process of the Video VAE is crucial, as we found that directly encoding sketches frame-by-frame as images yields suboptimal results.
>   *  **Step 2. Discard Empty Latents.** Latent frames corresponding to 4 purely blank sketches are discarded. We preserve only those latents that contain **at least one real sketch frame** within their 4-frame block to serve as conditions for injection.
>   *   **Step 3. Add Binary Mask.** Simultaneously, we generate a binary temporal mask along the channel dimension to indicate which specific frame(s) within the 4-frame block of each preserved latent contain valid content. This mask follows the official Wan implementation for transforming temporal masks into latent space. The combination of the preserved latent (containing empty frames) and this binary mask ensures the model remains aware of the precise keyframe location within a 4-frame block (i.e., within a single VAE-encoded latent).
>   *  **Step 4. Sketch Projection.** We then employ an additional patch embedding linear layer (trained separately) to transform these latents into the internal dimension of the DiT tokens. These sketch tokens are appended to the end of the original token sequence of the DiT backbone.
>   *   **Step 5. Positional Encoding Mapping.** During the attention operations of the DiT, we assign these sketch tokens the RoPE temporal indices that match the video tokens at the corresponding temporal positions.
>
>
> These steps ensure that the model correctly handles the VAE's temporal compression and maintains precise awareness of each sketch keyframe's temporal location. We have added this detailed description of our sparse sketch injection mechanism to Appendix A.3 of the revised paper.
>
> Moreover, we add more discussion and numerical results of the sketch–frame alignment evaluation. For each method, we run its output through the same sketch extractor used for creating synthetic sketches. At each keyframe index, we compare the extracted sketch to the input sketch using SSIM and the binarized pixel-level accuracy and report these numbers for all methods in a new table (shown below).
>
> ------
> | Model | Sketch SSIM↑ | Sketch Binary Accuracy↑ |
> |:------:|:----:|:-----:|
> | AniDoc | 0.5047 | 0.8196 |
> | LVCD | 0.5265 | 0.8347 |
> | ToonCrafter | 0.5278 | 0.8327 |
> | ToonComposer | **0.8360** | **0.9444** |
> ------
>
> This complements the existing metrics and directly measures adherence to the given keyframe sketch input. ToonComposer achieves significantly higher sketch fidelity compared to baselines, confirming precise temporal alignment. This is also reflected in Appendix C.8 of the revised paper.

---

> ### Author Response · Authors · 2025-11-28
> **Response to Reviewer pGDP (Part 2)**
>
> ### 3. **Suggestions and Minor issues**
>
> Thanks for the suggestions. We have moved the position-aware residual from the appendix into the main paper. We have also corrected the minor typos you noted.
>
> ### 4. **Region-wise control**
>
> We would like to clarify that our masks in region-wise control are not white-noise pixel masks. During training, we draw 3 types of connected geometric shapes (rectangles, circles, polygons) with random locations, sizes, aspect ratios, and coverage up to 60% of the frame. This encourages the model to fill in missing contiguous regions, rather than dealing with isolated pixels.
>
> For the training strategy, we chose random masks over semantic masks for two key reasons. The first reason is the alignment with user behavior. At inference time, users typically provide coarse masks that cover general areas rather than single, precise object segmentations. Geometric masks simulate this rough erasure more effectively than perfect semantic masks.
>
> The second reason is the generalization capability. Training on arbitrary geometric shapes encourages the model to develop a more robust contextual reasoning ability. It learns to generate plausible content based on surrounding textures and text prompts, regardless of the semantic category. In contrast, training solely on semantic masks might restrict the model to only filling specific, recognized object classes, reducing robustness to open-ended user inputs. Moreover, we empirically observe that the model generalizes well from geometric training masks to user-specified semantic regions (as in Figure 7 of the main paper).
>
> In addition to that, we have added more inference samples in Appendix C.7 to demonstrate this robustness (Figure 13 of the Appendix).

---

### Official Review · Reviewer_bpMj · 2025-10-30

**Soundness:** 3
**Presentation:** 3
**Contribution:** 2
**Rating:** 4
**Confidence:** 4

**Summary:**

In this paper, the authors claim that they are the first DiT-based cartoon and anime video production that unifies the traditionally separate inbetweening and colorization stages into a single "post-keyframing" process. They incorporate a sparse sketch injection mechanism for precise temporal and spatial control, and proposes a spatial low-rank adapter (SLRA) for effective domain adaptation to the cartoon style while preserving temporal priors. The system accepts as few as one keyframe sketch and one colored frame, reducing manual effort and allowing for flexible region-wise artist control. The authors curate a new dataset (PKData) for training and introduce PKBench, a benchmark with human-drawn sketches, to demonstrate ToonComposer's effectiveness. Experimental results show state-of-the-art performance compared to leading baselines in both synthetic and real-world sketch settings, with detailed ablations supporting each design choice.

**Strengths:**

1. The sparse sketch injection mechanism enables precise temporal motion control with minimal sketch input. The inclusion of region-wise control further increases usability and flexibility, allowing users to input incomplete sketches for targeted generation.
2. The introduction of the spatial low-rank adapter (SLRA) is a significant technical contribution. Ablation experiments show that SLRA consistently outperforms standard LoRA and other adaptation baselines by effectively tailoring only the spatial representation for the cartoon domain, thus keeping the video model’s temporal prior intact.
3.  The experiments are compelling: quantitative results on both synthetic and real-human benchmarks, extensive qualitative comparisons,  and in-depth ablations for each module. Meaningful discussions of motion consistency, aesthetics, and robustness to sketch styles are provided in the experiment section.

**Weaknesses:**

1. This is not the first DiT-based Anime in-betweening and colorization paper. SketchColour[1] and AnimeColor[2] were also based on DiT-based models. Though they may be concurrent works, the authors should still not claim that they are the first. Moreover, I personally don't recognize transferring a similar technique from UNet to DiT as a contribution.
2. The idea of only optimize spatial attention is not novel either. ToonCrafter[3] already found the fact that finetuning spatial layers only works well on anime in-betweening. Also, I don't recognize transferring a similar technique from UNet to DiT as a contribution.
3. From LayerAnimate Figure 1, I can see that they also have sparse sketches. The contribution of proposing sparse sketch seems not new either.

[1] Sadihin, B. C., Wang, M. H., Chua, S. P., & Su, H. (2025). SketchColour: Channel Concat Guided DiT-based Sketch-to-Colour Pipeline for 2D Animation. arXiv preprint arXiv:2507.01586.

[2] Zhang, Y., Wang, L., Wang, H., Wu, D., Lin, Z., Wang, F., & Song, L. (2025). AnimeColor: Reference-based Animation Colorization with Diffusion Transformers. arXiv preprint arXiv:2507.20158.

[3] Xing, J., Liu, H., Xia, M., Zhang, Y., Wang, X., Shan, Y., & Wong, T. T. (2024). Tooncrafter: Generative cartoon interpolation. ACM Transactions on Graphics (TOG), 43(6), 1-11.

[4] Yang, Y., Fan, L., Lin, Z., Wang, F., & Zhang, Z. (2025). LayerAnimate: Layer-level Control for Animation. arXiv preprint arXiv:2501.08295.

**Questions:**

The VAE of Wan2.1 has a temporal compression rate of 4. By using a sparse sketch, how do you process when only one or two sketches in successive 4 frames?

**Details Of Ethics Concerns:**

This research uses a large amount of anime data as datasets for training and benchmarking, which could potentially raise copyright issues.

---

> ### Author Response · Authors · 2025-11-28
> **Response to Reviewer bpMj (Part 1)**
>
> ### 1. **Novelty and "DiT-based" wording**
>
> We appreciate the reviewer pointing out concurrent works. We acknowledge that "DiT-based" is not the primary differentiator. Our core novelty lies in the **post-keyframing** paradigm, which fundamentally differs from existing works.
>
> Specifically, ToonComposer directly generates cartoon videos from very sparse control, typically one colored frame plus one or a few keyframe sketches, and avoids separate inbetweening and colorization pipelines.
>
> This design is motivated by how studio animators actually work: keyframes are created by senior artists, and the remaining frames and colorization are tedious and repetitive.
>
> We now emphasize this distinction more concretely by contrasting the input/output of these methods:
>
> ------
> | Method | Inputs | Outputs|Stage formulation|
> |:------:|:------:|:------:|:------:|
> | SketchColour  | Dense per-frame sketch, reference color frame | Colored frames | Colorization |
> | AnimeColor | Dense per-frame sketch, reference color frame | Colored frames | Colorization |
> | ToonCrafter | First and last frame | Inbetweened frames   | Inbetweening |
> | LayerAnimate | First/last frames with layer masks + dense per-frame control (e.g., sketch) | Cartoon video | Layer-level per-frame controlled generation (using dense sketches or trajectory) or inbeteweening |
> | **ToonComposer**  | First colored frame + **one or a few sparse keyframe sketches** | Cartoon video | Post-keyframing |
> ------
>
> This table clarifies that ToonComposer targets a different, more production-aligned stage and control regime.
>
> We have updated the introduction in our revised paper and have avoided using the "first DiT-based" wording to avoid misinterpretation.
>
>
> ###  2. **The spatial adaptation**
>
> We agree that the idea of directly training only the decoupled spatial components is not new and that ToonCrafter has shown in the UNet setting. Our contribution here is not simply "doing the same thing with DiT," but dealing with a different architectural situation. In ToonCrafter’s UNet, spatial and temporal processing are structurally decoupled. One can **literally** freeze temporal layers and tune spatial layers only. In the DiT video backbone we use, attention modules are fully spatio-temporal. Spatial and temporal dependencies are handled jointly inside each attention block. There is no clean spatial/temporal split that one can freeze or finetune independently.
>
> SLRA is designed specifically for this spatio-temporal-coupled setting and may extend to other backbones beyond DiT where full attention is used. It provides a spatial-only adaptation path in which native attention is fully spatio‑temporal, and it differs empirically from conventional LoRA adaptation.
>
> This design choice is supported by the ablations in Table 3 and Figure 8 of the main paper. With comparable or even lower parameter budgets, SLRA outperforms variants like LoRA.
>
>
> ### 3. **Sparse sketches and relation to LayerAnimate**
>
> We appreciate the reviewer pointing out the great recent work. We would like to clarify that LayerAnimate does not support the sparse sketch regime that ToonComposer targets for post-keyframing.
>
> Although Figure 1 of LayerAnimate looks like sparse control where an object occurs in later frames, this is due to the controlled layer not being present in the first several frames. It still requires per-frame sketches when that controlled object is presented in the scene.
>
> Also, LayerAnimate's "interpolation with sketches" setting, shown for example in Figure 4 of its paper, uses sketches on every frame and expects layer-wise control that covers all moving elements. The method is designed for layer-level control and interpolation, not for a post-keyframing scenario where an animator only draws one or two global keyframe sketches for an entire shot.
>
> In contrast, ToonComposer assumes extremely sparse keyframe sketches and a single colored reference frame.

---

> ### Author Response · Authors · 2025-11-28
> **Response to Reviewer bpMj (Part 2)**
>
> ### 4. **Details of temporal compression process for sketches**
>
> We thank the reviewer for the question regarding how we handle sparse sketches, given that Wan's VAE compresses 4 frames into a single latent frame. Our implementation proceeds as follows:
>   *  **Step 1. Encode with VAE.** We begin with sparse sketches in pixel space and construct a full-length sketch video by inserting all-zero (blank) sketch images at frames where no sketch is provided. We feed this sketch video through the same VAE encoder used for the RGB video (repeating the sketch across 3 RGB channels if necessary). This yields a sequence of latent frames, with one latent representing every 4 original frames (starting from the first frame). Maintaining the native encoding process of the Video VAE is crucial, as we found that directly encoding sketches frame-by-frame as images yields suboptimal results.
>   *  **Step 2. Discard Empty Latents.** Latent frames corresponding to 4 purely blank sketches are discarded. We preserve only those latents that contain **at least one real sketch frame** within their 4-frame block to serve as conditions for injection.
>   *   **Step 3. Add Binary Mask.** Simultaneously, we generate a binary temporal mask along the channel dimension to indicate which specific frame(s) within the 4-frame block of each preserved latent contain valid content. This mask follows the official Wan implementation for transforming temporal masks into latent space. The combination of the preserved latent (containing empty frames) and this binary mask ensures the model remains aware of the precise keyframe location within a 4-frame block (i.e., within a single VAE-encoded latent).
>   *  **Step 4. Sketch Projection.** We then employ an additional patch embedding linear layer (trained separately) to transform these latents into the internal dimension of the DiT tokens. These sketch tokens are appended to the end of the original token sequence of the DiT backbone.
>   *   **Step 5. Positional Encoding Mapping.** During the attention operations of the DiT, we assign these sketch tokens the RoPE temporal indices that match the video tokens at the corresponding temporal positions.
>
> These steps ensure that the model correctly handles the VAE's temporal compression and maintains precise awareness of each sketch keyframe's temporal location. We have added this detailed description of our sparse sketch injection mechanism to the Appendix of the revised paper.
>
> ### 5. **Ethics and copyright**
>
> PKData is constructed from professionally produced cartoon material from internal production and licensed sources. We have the right to use these clips for research and development, and we do not redistribute the raw video. Evaluation clips shown in the paper are used with explicit permission for testing and academic illustration.
>
> PKBench is created entirely in-house. All colored keyframes and sketches in PKBench are designed and drawn by professional artists commissioned for this project. PKBench is released for academic use with a license that restricts commercial reuse and requires attribution. We have clarified this in the Ethics Statement and dataset section.

---

### Author Response · Authors · 2025-11-28
**General Response**

We thank all the reviewers for their time and constructive comments.

We are encouraged that the reviewers **recognize the value** of our post-keyframing paradigm that unifies inbetweening and colorization to streamline animation production (Reviewers pGDP, ehPw, HHzB).

We appreciate the **positive feedback** on our technical contributions, specifically the sparse sketch injection mechanism for precise temporal keyframe control (Reviewers bpMj, ehPw) and the effective cartoon domain adaptation (Reviewers bpMj, pGDP).

Furthermore, reviewers acknowledged the contribution of our PKBench as valuable resources for the community (Reviewers pGDP, HHzB) and found our experimental results **compelling and strong** (Reviewers bpMj, pGDP, ehPw, HHzB).

Regarding the questions and concerns raised, we have addressed them in the point-by-point responses below. We have also revised the paper according to the insightful suggestions provided by the reviewers.

In addition, to fully appreciate the visual quality and the keyframe alignment fidelity of our method, we respectfully invite reviewers to view the **supplementary video** submitted previously if they have not done so already.

---

### Author Response · Authors · 2025-12-02
**Summary of Rebuttal (Part 1/2)**

Dear Area Chair,

We are aware of the recent OpenReview incident and the suspension of reviewer discussions. We understand this places a significant responsibility on the AC to evaluate the rebuttal materials directly. To assist in this process, we provide a concise summary of how we have systematically addressed the reviewers' concerns through detailed clarifications, new quantitative metrics, and additional comparative experiments.

We are encouraged that the reviewers acknowledged the value of our novel **post-keyframing paradigm** for cartoon production (Reviewers pGDP, ehPw, HHzB), the contribution of the **PKBench benchmark** featuring human-drawn sketches (Reviewers pGDP, HHzB), and the **compelling visual quality and strong results** of our method (Reviewers bpMj, pGDP, ehPw, HHzB). Below, we highlight how we have resolved the specific concerns regarding novelty, technical implementation, and evaluation.

### 1. Addressed Concerns on Novelty and Positioning (Reviewers bpMj, ehPw)
Some reviewers initially questioned the novelty or compared our work to concurrent methods. We addressed this by systematically differentiating our motivation from the limitations of existing pipelines:
- **Our Motivation & Differentiation:** Existing methods handle cartoon production stages separately (i.e., keyframing → inbetweening → colorization, see Figure 2 of the main paper). Previous ToonCrafter focuses solely on inbetweening, while methods like AniDoc, LVCD, SketchColour, and AnimeColor all require labor-intensive per-frame sketches for colorization (assuming inbetweening is already completed). Similarly, LayerAnimate relies on dense per-frame sketches for layer-wise control. This fragmentation leads to error accumulation and substantial manual effort.
Our ToonComposer bridges this gap by unifying inbetweening and colorization stages into **a novel post-keyframing stage**, generating complete cartoon videos from extremely sparse keyframe inputs (e.g., 1 sketch + 1 color frame). We have revised the introduction and provided a detailed feature comparison table in our response to Reviewer bpMj to clarify this scope.
- **Technical Contribution:** We emphasized that our contribution is the holistic pipeline tailored for this new **post-keyframing** task, driven by the sparse sketch injection, the spatial low-rank adapter (SLRA), and the region-wise control. Crucially, we clarified that SLRA is not merely "a low-rank adaptation on the spatial dimension" but a **tailored design for spatio-temporal coupled attention mechanisms**. In such architectures, conventional spatial-only adaptation (feasible in spatio-temporal decoupled methods like ToonCrafter) is structurally impossible. Our ablations (Table 3 and Figure 8 in the main paper) prove SLRA consistently outperforms standard LoRA in our task.

### 2. Resolved Technical Concerns on Sketch Control & Alignment (Reviewers bpMj, pGDP, ehPw)
Reviewers inquired about handling sparse sketches under the VAE's temporal compression, questioned alignment accuracy, and asked details about the region-wise control.
- **Implementation Details:** We provided a step-by-step explanation of our sparse sketch injection mechanism to clarify how we maintain precise temporal control despite VAE compression. This clarification has also been added to Appendix A.3.
- **Sketch Faithfulness/Alignment Evidence:** To address concerns about sketch faithfulness (Reviewer ehPw, pGDP), we introduced new metrics: *Sketch SSIM* and *Sketch Binary Accuracy* (higher are better). As detailed in the reply to Reviewer pGDP, ToonComposer achieves 0.8360 SSIM and 94.44% Accuracy, significantly outperforming baselines such as ToonCrafter (0.5278 SSIM, 83.27% Accuracy). This quantitatively proves our method's superior alignment (updated in Appendix C.8). Qualitative results showing the accuracy of keyframe sketch alignment are also provided in the supplementary video.
- **Region-wise Control:** We added a detailed discussion of the proposed region-wise control in the revised Appendix C.7, along with more samples showing its robustness when only a small area of keyframe sketches are provided.

---

> ### Author Response · Authors · 2025-12-02
> **Summary of Rebuttal (Part 2/2)**
>
> ### 3. Extensive New Experiments for Robustness and Generalization (Reviewer ehPw)
> Reviewer ehPw raised concerns regarding portability and requested comparisons with CtrlNet-like architectures, **explicitly stating a willingness to raise the score** if these were addressed. We have conducted **all requested experiments** and presented the results in our response to reviewer ehPw:
> - **Portability:** We successfully applied our method to the smaller Wan 2.1 1.3B model. The results demonstrate that our approach is model-agnostic and maintains high performance even without large-scale foundation models (Wan 2.1 14B).
> - **Comparison with CtrlNet:** We trained CtrlNet-based variants for both AnimateDiff and Wan 2.1 14B. ToonComposer significantly outperforms both baselines, proving the architectural superiority of our method over standard adapters.
> - **SLRA Ablations:** We completed the requested layer-placement sweep (early/mid/late) and rank ablations. These experiments confirm that our configuration offers the optimal performance (detailed in Appendix C.3 and C.4).
>
> ### 4. Clarifications on Dataset and Temporal Consistency (Reviewer HHzB)
> Reviewer HHzB (Score 8) supported acceptance and asked for data details.
> - **Dataset Details:** We clarified the composition of PKData and the distinction between synthetic training sketches and the human-drawn PKBench used for testing, proving the model's robustness to real-world artistic inputs.
> - **Temporal Metrics:** We added VBench motion consistency (higher is better) and FVD scores (lower is better) in our response, showing that SLRA (0.9906 consistency and 46.8062 FVD) effectively preserves the temporal priors of the foundation model compared to LoRA (0.9884 consistency and 47.2428 FVD).
>
> ### Conclusion
> We believe that our rebuttal comprehensively addresses every question raised. Given the initial support from Reviewer HHzB (Score 8) and pGDP (Score 6), along with the extensive new evidence addressing the concerns of Reviewers bpMj and ehPw, we are confident that ToonComposer represents a significant contribution to AI-assisted cartoon production. We respectfully request that the Area Chair consider these improvements in the final decision.
>
> Sincerely,
> Authors of Submission 18421

---

### Meta-Review · Area_Chair_xoAt · 2025-12-20

**Summary:**

The reviewers raised several key concerns that informed the evaluation of this submission:

- Novelty and Positioning (Reviewers bpMj, ehPw): Reviewers questioned the uniqueness of the work, noting concurrent DiT-based methods for animation (SketchColour, AnimeColor), prior art on spatial-only fine-tuning (ToonCrafter), and sparse control in other frameworks (LayerAnimate). The core claim of being "the first" was challenged.

- Technical Implementation & Faithfulness (Reviewers bpMj, pGDP, ehPw)

- Evaluation and Generalization (Reviewers ehPw, HHzB)

**Reviewer Concerns:**

- Novelty and Positioning: The authors successfully reframed their contribution away from being "the first DiT-based" method and toward the novel post-keyframing paradigm.

- Technical and Evaluation Clarifications are addressed.

**Reviewer Scores:**

- Reviewer bpMj (Initial: 4, Marginally Below): Their primary concerns (novelty claims, technical how, ethics) were directly and thoroughly addressed. The new positioning, detailed technical explanation, and dataset clarification would likely move their score upward to a 6 or 8 (Marginally Above/Acceptance).

- Reviewer pGDP (Initial: 6, Marginally Above): Their major concerns (temporal alignment proof, sketch faithfulness) were addressed with new quantitative metrics and detailed explanations. Their minor suggestions were accepted. They were already positive; this would solidify their score or potentially increase it.

- Reviewer ehPw (Initial: 2, Reject): They explicitly stated openness to changing their score based on responses. The authors conducted all requested experiments (CtrlNet comparisons, portability test, SLRA ablations) and added the requested metrics. This comprehensive address of major weaknesses would likely lead to a significant score increase to a 6 (Marginally Above).

- Reviewer HHzB (Initial: 8, Accept): Their supportive score would remain unchanged or be reinforced. Their requests for dataset details and temporal evidence were met.

---

### Decision · Program_Chairs · 2026-01-26

Accept (Poster)